# BRAINACTIV: IDENTIFYING VISUO-SEMANTIC PROPERTIES DRIVING CORTICAL SELECTIVITY USING DIFFUSION-BASED IMAGE MANIPULATION

**Diego García Cerdas**[*1]**, Christina Sartzetaki**[1]**, Magnus Petersen**[2]**,
Gemma Roig**[2]**, Pascal Mettes**[1]**, and Iris Groen**[1]
[1]University of Amsterdam, The Netherlands, [2]Goethe-Universität Frankfurt, Germany

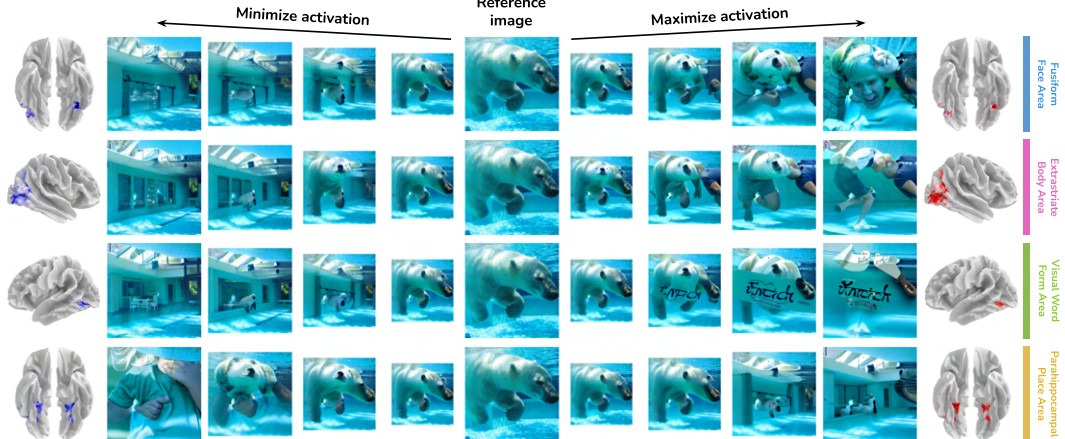

Figure 1: **BrainACTIV manipulates a reference image** to maximize or minimize the activity of a target region in the human visual cortex. By analyzing the resulting image variations, we can quantify visuo-semantic representations that underlie selective responses in the human brain. In the examples above, manipulations enhance hypothesized preferred categories in specific brain regions known to exhibit selectivity for faces, bodies, words, and places, respectively.

## ABSTRACT

The human brain efficiently represents visual inputs through specialized neural populations that selectively respond to specific categories. Advancements in generative modeling have enabled data-driven discovery of neural selectivity using brain-optimized image synthesis. However, current methods independently generate one sample at a time, without enforcing structural constraints on the generations; thus, these individual images have no explicit point of comparison, making it hard to discern which image features drive neural response selectivity. To address this issue, we introduce Brain Activation Control Through Image Variation (BrainACTIV), a method for manipulating a reference image to enhance or decrease activity in a target cortical region using pretrained diffusion models. Starting from a reference image allows for fine-grained and reliable offline identification of optimal visuo-semantic properties, as well as producing controlled stimuli for novel neuroimaging studies. We show that our manipulations effectively modulate predicted fMRI responses and agree with hypothesized preferred categories in established regions of interest, while remaining structurally close to the reference image. Moreover, we demonstrate how our method accentuates differences between brain regions that are selective to the same category, and how it could be used to explore neural representation of brain regions with unknown selectivities. Hence, BrainACTIV holds the potential to formulate robust hypotheses about brain representation and to facilitate the production of naturalistic stimuli for neuroscientific experiments.

*Corresponding author: diego.gcerdas@gmail.com

## 1 INTRODUCTION

The discovery of brain regions that selectively respond to specific image categories raises intriguing questions about their underlying neural representations (Grill-Spector and Weiner, 2014). While traditional approaches to measuring neural selectivity relied on a few hand-selected image categories, recent studies guide generative models with brain encoder gradients to activate category-selective regions of interest (ROIs) in human visual cortex (Ratan Murty et al., 2021; Ozcelik and VanRullen, 2023; Luo et al., 2023). These studies pioneered data-driven exploration of neural selectivity by optimizing random noise vectors to synthesize maximum-activating images, allowing the formulation of new hypotheses about the representations in each ROI. However, none of them explicitly enforce structural constraints on the generations; hence, the images are independently sampled without an explicit reference point. This process naturally leads to a varied set of images of which some characteristics are preferred by ROIs and others are randomly produced by the generative model. Disentangling these factors is essential for understanding the neural representations underlying category-selectivity and for determining the relative contribution of visual versus semantic features to neural representation, a key debate across scene (Groen et al., 2017), object (Bracci et al., 2017), face (Vinken et al., 2023) and word (Janini et al., 2022) perception.

We introduce *Brain Activation Control Through Image Variation* (BrainACTIV), a method for manipulating a reference image to increase or decrease predicted brain activity in a target cortical region, see Figure 1. BrainACTIV uses IP-Adapter (Ye et al., 2023) to prompt a pretrained diffusion model with brain-optimal embeddings obtained through spherical interpolation in CLIP space. Initial diffusion latents are computed with SDEdit (Meng et al., 2022) to retain the low-level structure of the reference image. The manipulation of a reference image (Goetschalckx et al., 2019; Papale et al., 2024) ensures a reliable comparison point for the synthesized stimuli, isolating the effect of brain optimality on the latter. Besides a straightforward visual interpretation, our method facilitates quantifying differences in visuo-semantic and mid-level image features using computer vision techniques, highlighting those preferred by a brain region of interest. Moreover, the use of a real image as reference enables the integration of BrainACTIV into novel hypothesis-driven studies.

We validate BrainACTIV by targeting fMRI responses in well-established category-selective ROIs, confirming that their predicted activation is successfully modulated by our image variations and that the visuo-semantic properties highlighted by them agree with previous neuroscientific work. Additionally, we demonstrate how our method can accentuate differences between similar regions of interest, providing insights into the specific role of each region in visual processing. Finally, we describe how researchers can select between semantic variation and low-level structural control when using BrainACTIV for experimental stimulus design. Our contributions are:

- The use of image manipulation to maximize or minimize responses in higher visual cortex: this guarantees that the changes made to the original image come from the objective of increasing or decreasing the brain activation, rather than stochasticity in the image generation process, with the original image serving for activity baseline comparison.
- The use of automated methods to quantify semantic category presence and mid-level image features to characterize each ROI in finer detail, circumventing the need for human behavioral assessments.
- The identification of differences in stimulus representation between similar brain regions beyond category selectivity, by accentuating these differences in a reference image.
- The introduction of BrainACTIV as a controllable method for neuroscientific stimulus generation, describing how researchers can modify the degree of low-level visual changes when generating image variations.

Our code is available at github.com/diegogcerdas/BrainACTIV.

## 2 RELATED WORK

**Category Selectivity in the Higher Visual Cortex.** Different regions in high-level areas of the human visual cortex exhibit selectivity for specific semantic categories like faces, bodies, places, and words (Kanwisher et al., 1997; McCarthy et al., 1997; Downing et al., 2001; Peelen and Downing, 2005; Epstein and Kanwisher, 1998; McCandliss et al., 2003). Reliable characterization of

each region requires measuring neural responses to large sets of images and finding those that elicit maximal activity (Ratan Murty et al., 2021). However, experimental constraints and the high dimensionality of image space make it impossible to test all potential stimuli. Neuroscientists have traditionally narrowed this search by focusing on hand-selected stimuli, but this risks overlooking relevant features that could not be conceived a priori. Deep neural networks (DNNs) trained on large-scale datasets of brain recordings have been adopted as "brain encoders" to make rapid and highly accurate predictions of neural responses to large volumes of images (Khosla et al., 2021). Moreover, deep generative models such as diffusion models Ho et al. (2020); Song et al. (2020) can synthesize novel stimuli by sampling from rich image priors constrained to the domain of natural images. Our work combines brain encoders and diffusion models to highlight semantic properties that drive functional selectivity in the visual cortex, enabling the formulation of new hypotheses more robustly and objectively than current data-driven approaches.

**Image Variation with Diffusion Models and CLIP.** Diffusion models treat the data generation process as iterative noise removal, progressively refining random noise $x_T \sim \mathcal{N}(0,1)$ into structured data $x_{T-1}, ..., x_{t+1}, x_t, x_{t-1}, ..., x_0$ through a trained denoising network. This process can be guided to synthesize samples from a conditional distribution, as done by text-to-image (T2I) models (Nichol et al., 2022; Saharia et al., 2022). Stable Diffusion (Rombach et al., 2022) enables efficient T2I synthesis by representing data in a lower-dimensional latent space. Image prompting allows for generating variations of a reference image $\mathcal{I}$, preserving its style and content. IP-Adapter Ye et al. (2023) introduces additional cross-attention layers in the denoising network of pretrained T2I models, incorporating information extracted by a CLIP image encoder (Radford et al., 2021). To preserve low-level structural fidelity to the reference image, SDEdit (Meng et al., 2022) initializes the denoising process at an intermediate step by injecting noise to $\mathcal{I}$ up to timestep $t_0 = \gamma \cdot T$ with $\gamma \in [0, 1]$ and using $x_{t_0}$ as starting point. Our work employs IP-Adapter and SDEdit on Stable Diffusion to generate image variations conditioned on brain-derived CLIP embeddings.

**Optimal Visual Stimulus Generation.** Previous studies have successfully used gradients from DNN-based brain encoders to produce stimuli that maximally activate parts of the macaque and mouse visual cortex (Bashivan et al., 2019; Walker et al., 2019; Ponce et al., 2019). Later approaches steered random noise vectors within generative models using encoder gradients to synthesize optimal stimuli for category-selective visual regions in macaques (Pierzchlewicz et al., 2024) and the human brain: NeuroGen (Gu et al., 2022) and Ratan Murty et al. (2021) used GANs (Goodfellow et al., 2014), while BrainDiVE (Luo et al., 2023) improved stimulus quality and semantic specificity by using diffusion models and a CLIP-based brain encoder. Diffusion-based generation has proven effective in "brain decoding" settings, where a visual stimulus is reconstructed based on elicited brain activation patterns (Chen et al., 2023; Scotti et al., 2023; Zeng et al., 2023; Ozcelik and VanRullen, 2023). In contrast, BrainDiVE and BrainACTIV synthesize novel stimuli that maximize predicted activity in specific brain regions. Because the noise vectors in BrainDiVE are randomly sampled for each synthesized image, this process leads to a varied stimulus set that shares some characteristics (i.e., those preferred by the region) and differs in others (i.e., those randomly produced by the generative model). This introduces the need for human behavioral studies to interpret large image sets to disentangle these features. Instead, our method of brain-targeted image variation ensures a point of comparison for each synthesized stimulus, directly disentangling the effect of brain activity optimality and allowing the quantification of semantic and mid-level image features relevant to the targeted cortical region using computer vision techniques. Concurrent work by Prince et al. (2024) explores the accentuation of pixel-based features in an image through gradient ascent to modulate brain activations; further, work by Papale et al. (2024) explores image perturbation through a GAN-based brain decoder (Dado et al., 2024) to study tuning properties of monkey IT neurons. In contrast, we leverage diffusion models and spherical interpolation in CLIP's latent space to study broader regions in the human visual cortex.

## 3 METHODS

Given a real reference image $\mathcal{I}$, we aim to produce variations highlighting semantic selectivity properties of a target cortical region. First, we explain how to condition diffusion models on a brain-derived signal to synthesize variations that increase or decrease predicted activations (Figure 2). Then, we describe how to quantify differences in semantic and mid-level image features to identify properties preferred by each region.

**1  Deriving modulation embeddings from brain encoder**

**2  Interpolating in CLIP space**

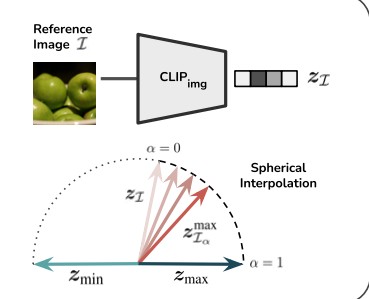

**3  Generating variations with diffusion model**

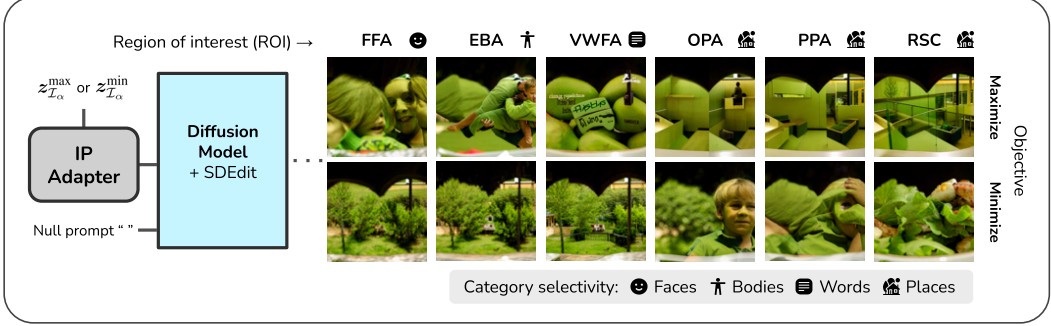

Figure 2: **Brain-targeted image variation pipeline**. **(1)** Modulation embeddings are derived from a CLIP-based brain encoder. **(2)** For a reference image, we produce intermediate embeddings using spherical interpolation in CLIP space. **(3)** An IP-Adapter conditions a pretrained diffusion model on the intermediate embeddings to generate images that maximize or minimize activity in category-selective ROIs; SDEdit helps retain low-level structural similarity to the reference image.

## 3.1  BRAIN-TARGETED IMAGE VARIATION

CLIP's semantically rich image embeddings have displayed high representational similarity to the higher visual cortex (Conwell et al., 2023; Wang et al., 2023), making them a suitable choice for representing and manipulating semantic content in the original image $\mathcal{I}$. Specifically, we move the image embedding $z_{\mathcal{I}} = \text{CLIP}_{\text{img}}(\mathcal{I})$ towards optimal endpoints that we derive from paired images and fMRI recordings. We refer to these endpoints as *modulation embeddings*.

First, similarly to BrainDiVE (Luo et al., 2023), we fit a brain encoder $f : \mathbb{R}^{H \times W \times 3} \rightarrow \mathbb{R}$ that transforms images $\mathcal{J}$ into brain activations $y$, where the latter are single values representing the average of voxel-wise beta values belonging to the region of interest (ROI)[1]. The brain encoder consists of two parts. The first is a frozen CLIP image encoder that outputs $D$-dimensional vectors. The second is a regularized linear regression model on normalized CLIP embeddings:

$$\left[ \frac{\text{CLIP}_{\text{img}}(\mathcal{J})}{\|\text{CLIP}_{\text{img}}(\mathcal{J})\|} \cdot \boldsymbol{w} + b \right] \Rightarrow y. \tag{1}$$

Due to the linear relationship between normalized embeddings and activations, $\boldsymbol{w} \in \mathbb{R}^D$ can be thought of as a vector in CLIP space that points in the direction of maximal activity for an ROI. Likewise, the negated weights $-\boldsymbol{w}$ point in the direction that minimizes it. Therefore, we define two ROI-specific modulation embeddings, $z_{\max}$ and $z_{\min}$, through the unit-norm weight vector:

$$z_{\max} = \frac{\boldsymbol{w}}{\|\boldsymbol{w}\|}, \quad z_{\min} = \frac{-\boldsymbol{w}}{\|\boldsymbol{w}\|}. \tag{2}$$

Luo et al. (2024) explain how to close the modality gap between CLIP embeddings of natural images and $z_{\max}$. First, for each image $M_i$ in a set of $K$ natural images $\mathbb{M} = \{M_1, M_2, \cdots, M_K\}$, a

---

[1]While we use ROI-wise averaged brain responses, this method could be straightforwardly adapted to smaller cortical regions or even single voxels.

softmax score with temperature $\tau$ is computed through

$$\text{score}_i = \frac{\exp(S_{\cos}(z_{\max}, e_i)/\tau)}{\sum_{k=1}^{K} \exp(S_{\cos}(z_{\max}, e_k)/\tau)}, \tag{3}$$

where $e_i = \text{CLIP}_{\text{img}}(M_i)$ and $S_{\cos}(\cdot, \cdot)$ is the cosine similarity function. Then, $z_{\max}$ is projected to the space of CLIP embeddings for natural images through a decoupled weighted sum of the image embeddings:

$$z_{\max}^{\text{proj}} = \left( \sum_{k=1}^{K} \text{score}_k \cdot \|e_k\| \right) \cdot \left( \sum_{k=1}^{K} \text{score}_k \cdot \frac{e_k}{\|e_k\|} \right). \tag{4}$$

A similar procedure can be followed for $z_{\min}$. In the following, we assume both $z_{\max}$ and $z_{\min}$ are projected unless otherwise stated.

Next, we use modulation embedding $z_{\max}$ and the reference image embedding $z_{\mathcal{I}}$ to produce *intermediate embeddings* $z_{\mathcal{I}_\alpha}^{\max}$ using spherical linear interpolation:

$$z_{\mathcal{I}_\alpha}^{\max} = \frac{\sin((1-\alpha) \cdot \theta)}{\sin(\theta)} z_{\mathcal{I}} + \frac{\sin(\alpha \cdot \theta)}{\sin(\theta)} z_{\max}', \tag{5}$$

where $\theta = \cos^{-1}\left(\frac{z_{\mathcal{I}} \cdot z_{\max}'}{\|z_{\mathcal{I}}\| \cdot \|z_{\max}'\|}\right)$ is the angle between the vectors, $\alpha \in [0, 1]$ indicates the extent of rotation, and $z_{\max}' = \|z_{\mathcal{I}}\| \cdot z_{\max}$. Larger values of $\alpha$ are thus expected to increase activations in the target ROI. An analogous operation with $z_{\min}$ yields intermediate embeddings $z_{\mathcal{I}_\alpha}^{\min}$.

Finally, we perform guided image synthesis with Stable Diffusion (Rombach et al., 2022) to generate the image variations $\mathcal{I}_\alpha$. To incorporate the brain-optimized semantic information from $\mathcal{I}$ into $\mathcal{I}_\alpha$, we use an IP-Adapter (Ye et al., 2023) to prompt the diffusion model with $z_{\mathcal{I}_\alpha}^{\max}$ or $z_{\mathcal{I}_\alpha}^{\min}$ (skipping the adapter's prepended image encoder). To retain the low-level structure of $\mathcal{I}$ in $\mathcal{I}_\alpha$, we obtain the initial diffusion latents $x_{t_0}$ through SDEdit with $t_0 = \gamma \cdot T$, where $T$ is the total number of denoising steps and $\gamma \in [0, 1]$. The hyperparameters $\alpha$ and $\gamma$ specify the degree of semantic variation and structural control in the manipulations (subsection 4.5).

## 3.2 QUANTIFYING INFORMATION IN ACTIVITY-MAXIMIZING AND MINIMIZING IMAGES

We identify the effect of brain optimization in the image variations $\mathcal{I}_\alpha$ by quantifying differences in category presence and mid-level image features with respect to the reference $\mathcal{I}$. We focus on 16 categories based on previous research on cortical representation and behavioral judgments (Huth et al., 2012; King et al., 2019; Hebart et al., 2020): *faces*, *hands*, *feet*, *people*, *animals*, *plants*, *food*, *furniture*, *tools*, *clothing*, *electronics*, *vehicles*, *landscapes*, *buildings*, *rooms*, and *text*. For each category, we build a representation embedding with CLIP. A challenge in doing so is that single-word descriptions are typically insufficient to capture all possible category instances. Therefore, we build the embeddings using an overcomplete set of concrete nouns from WordNet (Miller, 1995) classified by a large language model (details in appendix subsection A.1). Hence, we measure category presence through cosine similarity between an image embedding and the category's embedding.

To illustrate how BrainACTIV can reveal not only high-level categorical, but also low-level structural changes in brain-optimized images, we compute a number of mid-level features: *entropy*, the minimum number of bits needed to encode the gray level distribution in a local neighborhood, as a loose quantification of texture/clutter, which is known to affect many aspects of human vision Rosenholtz et al. (2007); and inspired by prior work showing that metrics of 3D scene structure are represented in scene-selective ROIs (Lescroart and Gallant, 2019; Dwivedi et al., 2021; Sarch et al., 2023), we also computed *metric depth*, estimated with the ZoeDepth network (Bhat et al., 2023), and *Gaussian curvature* and *surface normals*, computed with the XTC network (examples for reference NSD images can be found in Appendix subsection A.11).

# 4 RESULTS

## 4.1 SETUP

We use the Natural Scenes Dataset (NSD) (Allen et al., 2022), a large dataset of whole-brain high-resolution fMRI responses from eight human subjects. Each subject viewed ∼10,000 nat-

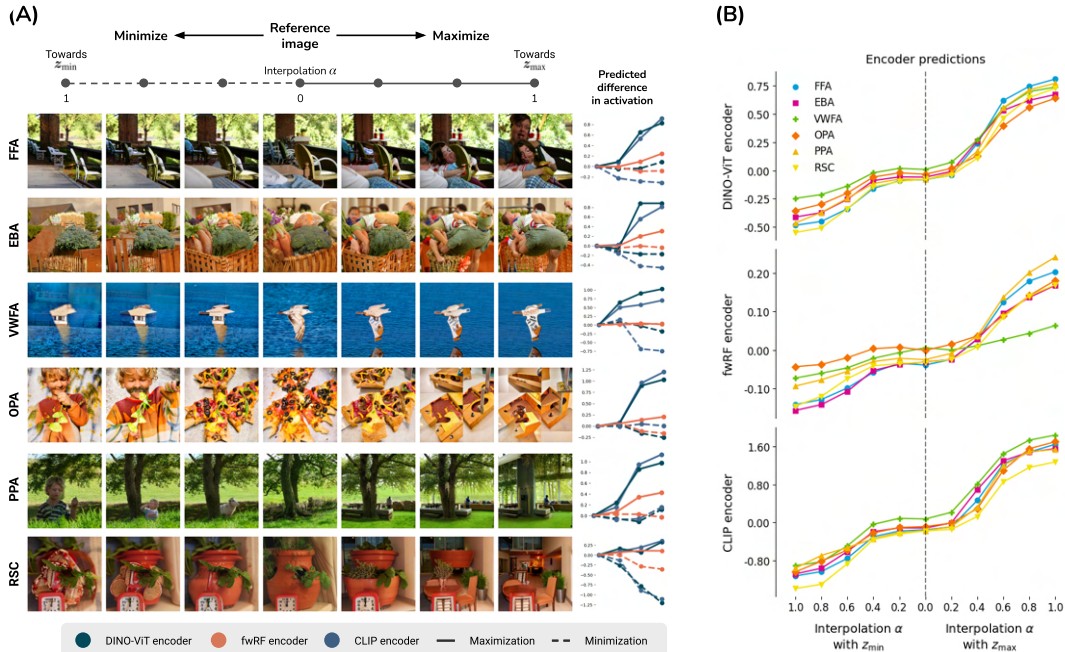

Figure 3: **Example variations and modulation results**. **(A)** Example image variations show the effect of activation maximization and minimization in each ROI; line plots show predicted differences in activation with respect to the reference image. **(B)** ROI activations predicted by DINO-ViT encoder (top), fwRF encoder (middle), and CLIP encoder (bottom) as a function of interpolation $\alpha$ with each modulation embedding, averaged across test images and subjects.

ural images from the MS COCO dataset (Lin et al., 2014) repeated three times across multiple scanning sessions. The fMRI beta values are z-scored within their original session and averaged across repetitions. Following standard practice in fMRI encoding (van Gerven, 2017; Naselaris et al., 2011), we split the data into a shared test set consisting of the 1,000 images seen by all subjects and a subject-specific training set with the remaining images. The training sets are used to analytically derive weights $w$ during modulation embedding derivation, as well as for embedding projection (using $\tau = 0.01$). We use a projection set consisting of 400,000 images from the `laion/relaion2B-en-research-safe` dataset (Schuhmann et al., 2022). We use the functional localizer masks included in NSD (thresholding at $t > 5$) to define cortical regions of interest.

We employ a pretrained Stable Diffusion model (`stable-diffusion-v1-5`) (Rombach et al., 2022) for guided image synthesis and a pretrained IP-Adapter (`ip-adapter_sd15`) (Ye et al., 2023) for image embedding conditioning. For consistency with these models, we use Open-CLIP's `ViT-H/14` CLIP architecture with `LAION2B-S32B-B79K` pretrained weights (Radford et al., 2021; Ilharco et al., 2021; Schuhmann et al., 2022). We use a separate brain encoder to predict activations in our experiments. Its architecture consists of DINOv2 (Oquab et al., 2023) as a feature extractor, followed by an ensemble of single-layer vision transformers (ViTs) (Dosovitskiy et al., 2021) and multilayer perceptrons, inspired by Adeli et al. (2023). To ensure the robustness of our method, we employ an additional feature-weighted receptive field encoder (St-Yves and Naselaris, 2018; Allen et al., 2022) (available through the Neural Encoding Dataset (Gifford and Cichy, 2024)) in our validation procedure. We refer to these as *DINO-ViT encoder* and *fwRF encoder*, respectively. Importantly, neither encoder is CLIP-based; hence, they do not share the same latent space as the encoders used to derive the modulation embeddings. Details on architecture and prediction performance of all encoders can be found in the appendix subsection A.2.

## 4.2 MODULATING ACTIVITY IN BRAIN REGIONS OF INTEREST

We validate BrainACTIV by targeting six previously identified regions of interest in the higher visual cortex: fusiform face area (FFA), extrastriate body area (EBA), visual word form area (VWFA), occipital place area (OPA), parahippocampal place area (PPA), and retrosplenial cortex (RSC) (pre-

| ROI | $L_2$ ($\downarrow$) | | | LPIPS ($\downarrow$) | | |
| :---: | :---: | :---: | :---: | :---: | :---: | :---: |
| | Random | Maximize | Minimize | Random | Maximize | Minimize |
| FFA | | $31.9_{\pm 4.9}$ | $30.1_{\pm 5.4}$ | | $0.27_{\pm 0.09}$ | $0.27_{\pm 0.09}$ |
| EBA | | $30.7_{\pm 5.8}$ | $30.8_{\pm 5.2}$ | | $0.24_{\pm 0.09}$ | $0.28_{\pm 0.10}$ |
| VWFA | $79.2_{\pm 15.7}$ | $29.9_{\pm 4.8}$ | $30.5_{\pm 4.7}$ | $0.55_{\pm 0.09}$ | $0.26_{\pm 0.10}$ | $0.28_{\pm 0.10}$ |
| OPA | | $31.3_{\pm 4.7}$ | $29.9_{\pm 5.5}$ | | $0.29_{\pm 0.11}$ | $0.24_{\pm 0.09}$ |
| PPA | | $32.3_{\pm 4.7}$ | $29.4_{\pm 5.6}$ | | $0.31_{\pm 0.11}$ | $0.26_{\pm 0.10}$ |
| RSC | | $31.1_{\pm 4.9}$ | $32.1_{\pm 4.1}$ | | $0.27_{\pm 0.10}$ | $0.29_{\pm 0.10}$ |

Table 1: **Structural control metrics**. Image variations remain structurally similar to the reference image even at $\alpha = 1$ for maximization and minimization objectives; hence, they serve as a reliable comparison point to quantify preferred features. Additional baselines in Appendix subsection A.12.

cise location can be found in Allen et al. (2022)). First, we identify six mutually exclusive subsets of images in NSD that share broadly similar semantic contexts: *wild animals*, *birds*, *vehicles*, *people in sports*, *food*, and *furniture*. We define each subset by filtering the pixel-wise category annotations made available with COCO (appendix subsection A.3). These subsets are employed to enforce the diversity of image selection in our experiment.

For each subject and each ROI, we select the 20 test images from each subset with measured responses closest to baseline activation (i.e., the average ROI activation across all test images). Because initial experiments showed that modulation embeddings are highly similar across subjects (see Appendix subsection A.9), we opt to use the average of all subject-specific $z_{\max}$ and $z_{\min}$ (before projection), excluding the subject on which predictions are made. Hence, we are modulating brain activity in each subject through a signal (averaged $z_{\max}$ or $z_{\min}$) derived exclusively from the rest of the subjects' data. We manipulate each of the 120 selected test images with interpolation values $\alpha \in \{0.1, 0.2, ..., 0.9, 1\}$, producing 20 variations in total for each image. For SDEdit, we use a logarithmic warm-up schedule for $\gamma$ up to a value of $\gamma = 0.6$ for $\alpha = 1$. Note that $\alpha = 0$ corresponds to the unaltered test image—the diffusion model is not used. Next, we predict activations for each of them using the appropriate subject- and ROI-specific DINO-ViT encoder and fwRF encoder. To compute the predicted *difference* in activation for each variation, we subtract the prediction of the reference image.

Figure 3 (A) displays example variations for each ROI, along with the predicted differences in activation. Note that the effect of $\alpha$ on the magnitude of these differences varies for each image, as an effect of its features and the ROI's sensitivity to these. To study a region's selectivity, we look for features that consistently appear or disappear over a wide range of contexts. Thus, our analyses focus on the general effect of BrainACTIV over the whole selection of test images. Additional examples can be found in the appendix subsection A.4.

First, we verify that the reference images serve as a reliable comparison point by measuring how structurally similar they are to the variations. Following Meng et al. (2022), we compute image $L_2$ distance and LPIPS (Zhang et al., 2018) between reference and variations, averaged over images and subjects. As a baseline, we compare 1,000 random pairs in the test set. Table 1 shows that structural similarity is preserved on maximization and minimization ($\alpha = 1$) for all ROIs.

Next, we look at the effect of our variations on the DINO-ViT and fwRF encoder outputs to verify that BrainACTIV successfully increases and decreases predicted ROI responses. Figure 3 (B) shows these predictions as a function of $\alpha$ for all ROIs (averaged over images and subjects). We observe a stable increase and decrease across ROIs for both encoders, confirming that our method modulates predicted activations. The plots show an expected lag in activity increase/decrease up to $\alpha \approx 0.4$ due to our $\gamma$ schedule since we intended the initial variations to be close to the reference image. Furthermore, we observe a similar effect with the CLIP encoder used to manipulate the images.

### 4.3 QUANTIFYING VISUO-SEMANTIC CHANGES IN IMAGE VARIATIONS

In this section, we verify that the category selectivity suggested by BrainACTIV for each ROI agrees with established neuroscientific findings. To this end, we use the manipulations from subsection 4.2 to identify the categories whose presence is increased when activations are maximized (Figure 4, top row). The plots display differences in category presence (relative to the reference) averaged over

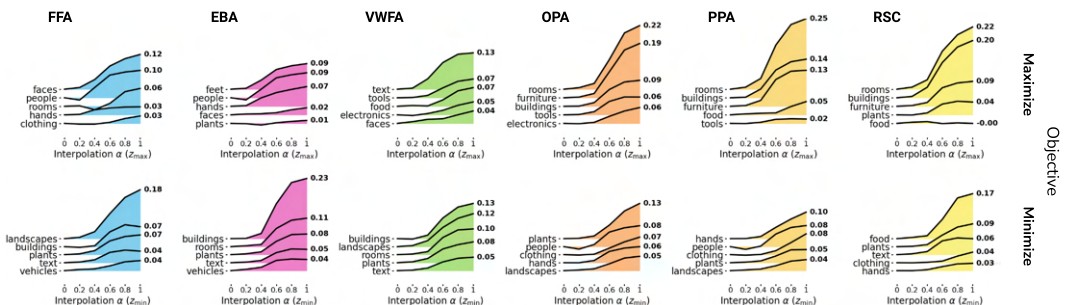

Figure 4: **Top categories for each region**. Each plot displays the difference in category presence (see subsection 3.2) with respect to the reference image as a function of $\alpha$. Top-5 categories per ROI for maximization (top) and minimization objective (bottom) are ranked by the highest measured difference. Results agree with hypothesized preferred categories.

all images and subjects. FFA increases the presence of faces, agreeing with Kanwisher et al. (1997) and McCarthy et al. (1997); EBA increases body parts (Downing et al., 2001); VWFA increases text (McCandliss et al., 2003) and tools/food, potentially suggesting preference for text on small objects [2]; OPA, PPA, and RSC increase manmade structures/scenes (Kamps et al., 2016; Epstein and Kanwisher, 1998; Mitchell et al., 2018). Category presence also increased during activation minimization (Figure 4, bottom row): FFA, EBA, and VWFA respond minimally for scenes/structures (particularly for FFA, landscapes), OPA and PPA to people/plants, and RSC to food. These minimizations are also broadly consistent with existing literature: the opposite preference of face- versus place-selective regions is commonly observed in fMRI (e.g. Silson et al., 2022; Margalit et al., 2020), and the minimal preference for plants in place-regions could reflect a preference for built/man-made structure (Çukur et al., 2016; Groen et al., 2021). However, others are novel; e.g. a minimal preference for food in RSC has, to our knowledge, not been reported before.

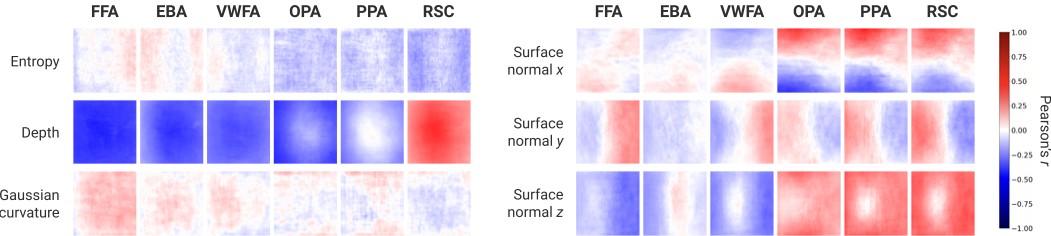

Figure 5: **Mid-level features**. Correlation between predicted ROI activation differences and mid-level feature differences at each pixel location in an image.

Because BrainACTIV allows the generation of image variations over a wide range of activation values, we can use it to study the correlation between predicted activation differences and mid-level feature differences for different locations in an image (Figure 5). Results suggest an important role of surface orientation in differentiating scene- and object-selective regions, with FFA, EBA, and VWFA preferring surfaces pointing outwards and OPA, PPA, and RSC preferring surfaces pointing inward, as reported before in controlled stimulus sets (Cheng et al., 2021). Moreover, the enhanced correlations with surface normals in scene-selective regions are consistent with their reported sensitivity to 3D configurations (Lescroart and Gallant, 2019). Depth correlations further emphasize differences between scene-selective regions, with RSC showing a higher correlation with deeper depth values. This could potentially reflect a role for RSC in coding perceived egocentric distances (Persichetti and Dilks, 2016). Together with section 4.2, these results demonstrate the validity of BrainACTIV as a data-driven method that reproduces known properties of visual cortex and can help formulate fine-grained new hypotheses about image properties driving brain activations.

---

[2] It is important to note that the projection of modulation embeddings to the space of CLIP image embeddings necessarily biases the representativity of particular features towards objects in the projection set that most frequently hold these features (e.g., small size → food).

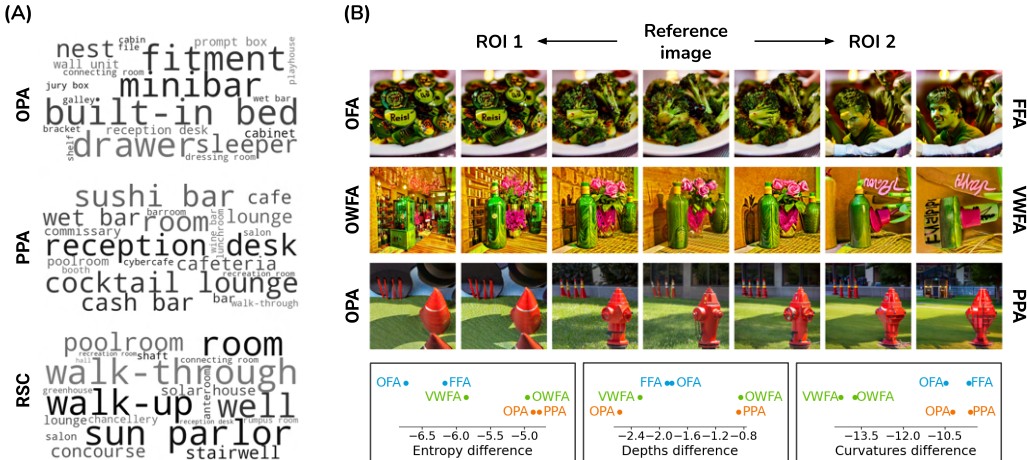

Figure 6: **Differences between similar ROIs**. **(A)** Top nouns emphasize differences between scene-selective ROIs. **(B)** (Top) Example variations accentuating differences for three ROI pairs: OFA-FFA, OWFA-VWFA, OPA-PPA. (Bottom) Differences in mid-level feature values (with respect to reference image) of each ROI, averaged and z-scored over test images.

## 4.4 IDENTIFYING DIFFERENCES BETWEEN SIMILAR ROIS

BrainACTIV can also be used to identify what distinguishes one region from another beyond category selectivity, an important step toward understanding broader functional organization principles in the visual cortex. We perform a top-nouns analysis (Figure 6 (A)) to identify the WordNet nouns whose presence increases the most on our maximization results for subsection 4.2. This analysis already highlights differences between the three scene-selective regions: OPA prefers local scene elements, while PPA and RSC prefer more global views (Kamps et al., 2016; Henderson et al., 2008); additionally, RSC prefers corridor-like scenes, potentially related to its role in navigation (Mitchell et al., 2018). These results further demonstrate how BrainACTIV improves upon BrainDiVE and NeuroGen while retaining their fine-grained distinction capabilities.

However, BrainACTIV can also be adapted to directly generate new hypotheses about ROIs with similar category-selectivity, through accentuation of differences between ROIs by manipulating a reference image. To demonstrate this, we here target three pairs of ROIs that are selective to the same category: face-selective OFA and FFA, word-selective OWFA and VWFA, and place-selective OPA and PPA. For each pair, we create *accentuation embeddings* by subtracting the modulation embeddings of each ROI from one another. We randomly sample 50 images from the test set and manipulate each of them with interpolation values $\alpha \in \{0.1, 0.2, ..., 0.9, 1\}$ toward the accentuation embeddings. For SDEdit, we use an exponential warm-up schedule up to $\gamma = 0.6$ for $\alpha = 1$. Figure 6 (B) displays example variations for each ROI pair and measured differences in entropy, depth, and curvature. Additional examples can be found in Appendix subsection A.5. The features accentuated on each side represent preferred visual properties that distinguish the regions.

BrainACTIV suggests a higher preference for text in OFA and a higher preference for faces in FFA, despite both being face-selective. For OWFA, we identify a higher preference for cluttered coarse-grained elements, evidenced by higher entropy values; VWFA shows a preference for text on small items. Finally, OPA and PPA differ in sensitivity to depth as analyzed in subsection 4.2. These new hypotheses can be validated by using these images as experimental stimuli in new fMRI studies.

## 4.5 PRODUCING NOVEL EXPERIMENTAL STIMULI

BrainACTIV generates synthetic stimuli that differ from a real reference image along a hypothesized tuning axis—derived in a data-driven manner—for a particular ROI. These paired images can be employed as stimuli for novel neuroscientific experiments (Figure 7 (A)). To facilitate its use for researchers and illustrate the available design choices, we briefly show the effect of our two hyperparameters—interpolation $\alpha$ and SDEdit $\gamma$—on the resulting images (Figure 7 (B). Both hyperparameters decrease semantic similarity and structural fidelity to the reference image (as evi-

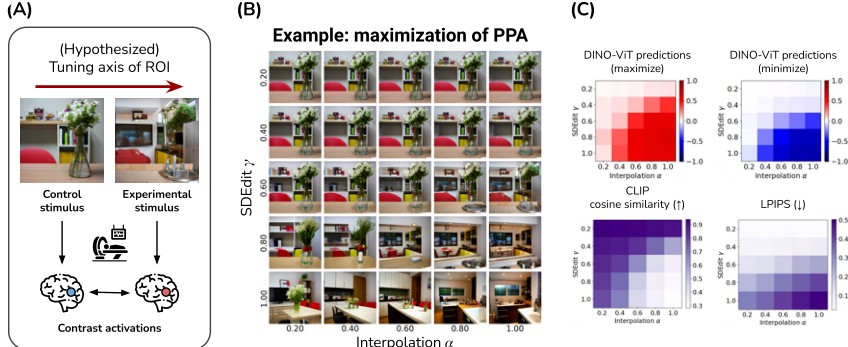

Figure 7: **Effect of $\alpha$ and $\gamma$ for novel stimuli**. **(A)** Schematic of the use of BrainACTIV on neuroscientific experiments. **(B)** Different values of $\alpha$ and $\gamma$ present a choice between structural fidelity and semantic variation. **(C)** Predicted activations are more strongly modulated when semantic content is closer to $z_{\max}$ or $z_{\min}$; lower $\alpha$ and $\gamma$ result in higher semantic similarity and structural fidelity to the reference image.

denced by lower cosine similarities and higher LPIPS metrics, respectively) (Figure 7 (C)). At the same time, we observe that lower semantic similarity and lower structural fidelity result in higher changes in predicted activations. This is to be expected from the design of our modulation embeddings. However, two distinct alternatives exist: choosing lower $\alpha$ and higher $\gamma$ results in variations that mostly retain the semantic content of the reference image while the spatial arrangement differs (depending on how well CLIP can capture it). Conversely, higher $\alpha$ and lower $\gamma$ favor the low-level structure of the reference image while more strongly varying the semantic content.

## 5 DISCUSSION

We introduced BrainACTIV, a method for modulating predicted brain responses through image manipulation. To our knowledge, we are the first work to use generative models—particularly, diffusion models—to manipulate reference images with the goal of maximizing or minimizing activations in the human visual cortex. Our results show the potential of our approach for fine-grained and reliable identification of visuo-semantic properties preferred by specialized neural populations. This information can be used to formulate new hypotheses about visual representations in the brain.

We propose that our generative framework can be employed by neuroscientists to design precisely controlled and innovative experimental paradigms to disambiguate the role of low-, mid- and high-level features, whose inherent correlations in natural images complicates the ability to isolate their effect on brain responses (Malcolm et al., 2016; Lescroart et al., 2015). We here primarily demonstrate our approach on brain regions with known category-selectivity, but also explore earlier visual regions and anterior IT (see Appendix subsection A.6 and subsection A.7) to highlight how BrainACTIV could help interpret 'no-mans land' regions of cortex (Bao et al., 2020) whose functional specialization is less well understood. Future work can explore BrainACTIV's manipulation framework in alternative stimulus modalities, such as natural language (Luo et al., 2024; Tuckute et al., 2024). Finally, given BrainACTIVs reduced computational need relative to prior work, we believe it holds potential for exploration of selectivity in closed-loop paradigms where activations are continuously updated along an optimization trajectory (e.g. Ponce et al., 2019).

Our method has some limitations. First, because we employ pretrained models for image synthesis and representation, our results are subjected to biases in their training data. These biases might over-represent certain categories through correlations with specific image features, producing misleading results. We encourage future work to use fine-tuned models and domain-specific representation spaces to explore finer-grained selectivity within smaller specialized cortical regions. Second, our work relies on brain encoders to validate the effective modulation of brain activity. While we have taken measures to ensure the robustness of our results, future work should validate BrainACTIV's predicted activations against novel brain recordings. In summary, BrainACTIV unlocks the possibility to test existing and generate novel hypotheses about neural representations in visual cortex using brain-guided image diffusion with structural control.

## 6 ACKNOWLEDGMENTS

The presentation of this paper at the conference was financially supported by the Amsterdam ELLIS Unit (European Laboratory for Learning and Intelligent Systems).

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

# A APPENDIX

## A.1 CATEGORY REPRESENTATIONS IN CLIP

We build category-specific CLIP embeddings to quantify the presence of each category (*faces*, *hands*, *feet*, *people*, *animals*, *plants*, *food*, *furniture*, *tools*, *clothing*, *electronics*, *vehicles*, *landscapes*, *buildings*, *rooms*, and *text*) in an image through cosine similarity with the image embedding. First, we gather a large list of $N_{\text{nouns}} = 17,086$ concrete nouns from WordNet (Miller, 1995) by collecting all hyponyms of the following synsets:

- `amphibian.n.03`
- `article.n.02`
- `bird.n.01`
- `body_of_water.n.01`
- `building.n.01`
- `commodity.n.01`
- `correspondence.n.01`
- `external_body_part.n.01`
- `facility.n.01`
- `fish.n.01`
- `food.n.02`
- `instrumentality.n.03`
- `land.n.04`
- `person.n.01`
- `placental.n.01`
- `plant.n.02`
- `plaything.n.01`
- `geological_formation.n.01`
- `publication.n.01`
- `reptile.n.01`
- `room.n.01`
- `sign.n.02`
- `vehicle.n.01`
- `way.n.06`
- `written_record.n.01`

Then, we perform zero-shot classification of each noun into one of the categories using `facebook/bart-large-mnli`, a version of the language model BART (Lewis et al., 2020) trained on the MultiNLI dataset (Williams et al., 2018; Yin et al., 2019). Instead of using the category names as labels, we build custom labels:

- faces: "related to faces, eyes, nose, mouth"
- hands: "related to hands, arms, fingers"
- feet: "related to feet, legs, toes"
- people: "related to people, humans, persons"
- animals: "related to animals, creatures, fauna"
- plants: "related to plants, greenery, flora"
- food: "related to food, meals, eating"
- furniture: "related to furniture, household items"
- tools: "related to tools, equipment, instruments"
- clothing: "related to clothing, textiles, garments"
- electronics: "related to electronics, gadgets, devices"
- vehicles: "related to vehicles, transportation, travel"
- landscapes: "related to natural areas, landscapes, outdoors"
- buildings: "related to urban areas, buildings, structures"
- rooms: "related to indoors, rooms, interiors"
- text: "related to written text, signs"

The resulting class probabilities are gathered in a matrix $\boldsymbol{Y}_{\text{prob}} \in [0,1]^{N_{\text{nouns}} \times 16}$ where each row sums up to 1. We weigh the probabilities by the salience of each category with respect to the rest for each noun to obtain $\boldsymbol{Y}_{\text{sal}}$:

$$[\boldsymbol{Y}_{\text{sal}}]_{i,j} = [\boldsymbol{Y}_{\text{prob}}]_{i,j} \cdot \frac{[\boldsymbol{Y}_{\text{prob}}]_{i,j}}{\sum_k [\boldsymbol{Y}_{\text{prob}}]_{i,k}}.$$

Next, we compute embeddings for each of the nouns using CLIP's text encoder. To make these more robust, we average the embeddings obtained through 18 prompt templates (e.g., *"a photo of a {}."* or *"a good photo of the {}."*) used originally by CLIP for image classification (Radford et al., 2021). We normalize these embeddings and gather them in a matrix $\boldsymbol{Z}_{\text{nouns}} \in \mathbb{R}^{N_{\text{nouns}} \times 1024}$.

Finally, we use a regularized linear regression model on $\boldsymbol{Z}_{\text{nouns}}$ to predict $\boldsymbol{Y}_{\text{sal}}$ and analytically derive the weights $\boldsymbol{W}_{\text{nouns}} \in \mathbb{R}^{D \times 16}$. Each column in the weight matrix then functions as our representation for each category. We notice that the *text* category is difficult to represent through this method; therefore, we instead compute its embedding by encoding the phrase "text on an object" using each of the 18 prompt templates and averaging them. Figure 8 displays each category's representative examples from the Natural Scenes Dataset (NSD) Allen et al. (2022), as well as salient WordNet nouns.

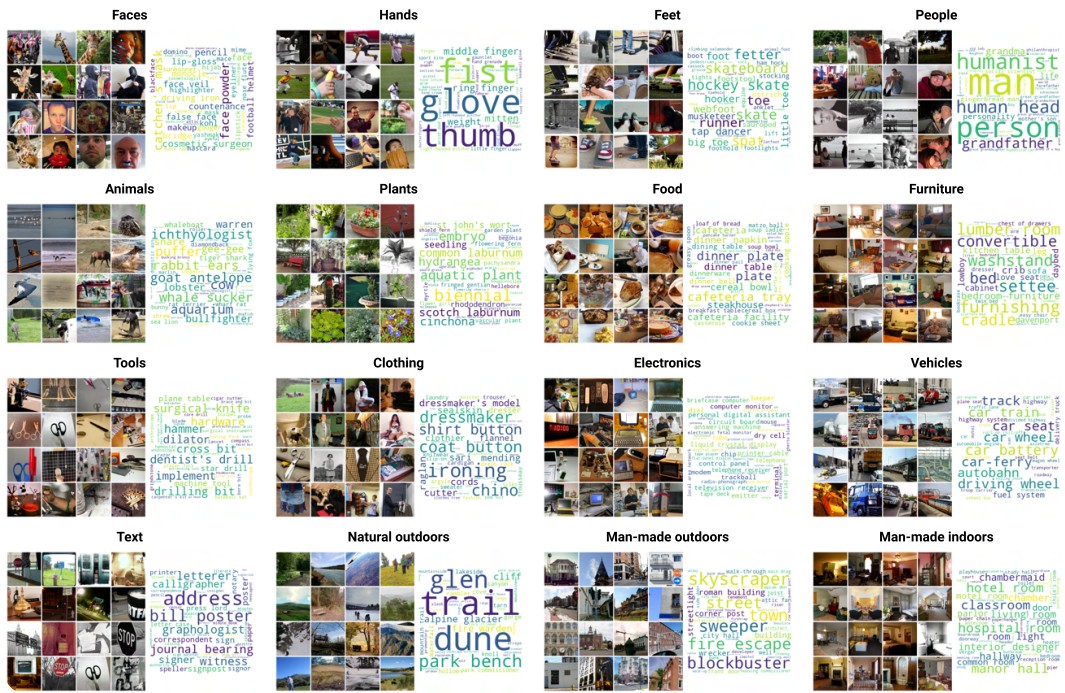

Figure 8: Representative examples (high cosine similarity) from NSD for each category, together with top WordNet nouns as classified by `facebook/bart-large-mnli`.

## A.2   BRAIN ENCODER PERFORMANCE AND DETAILS

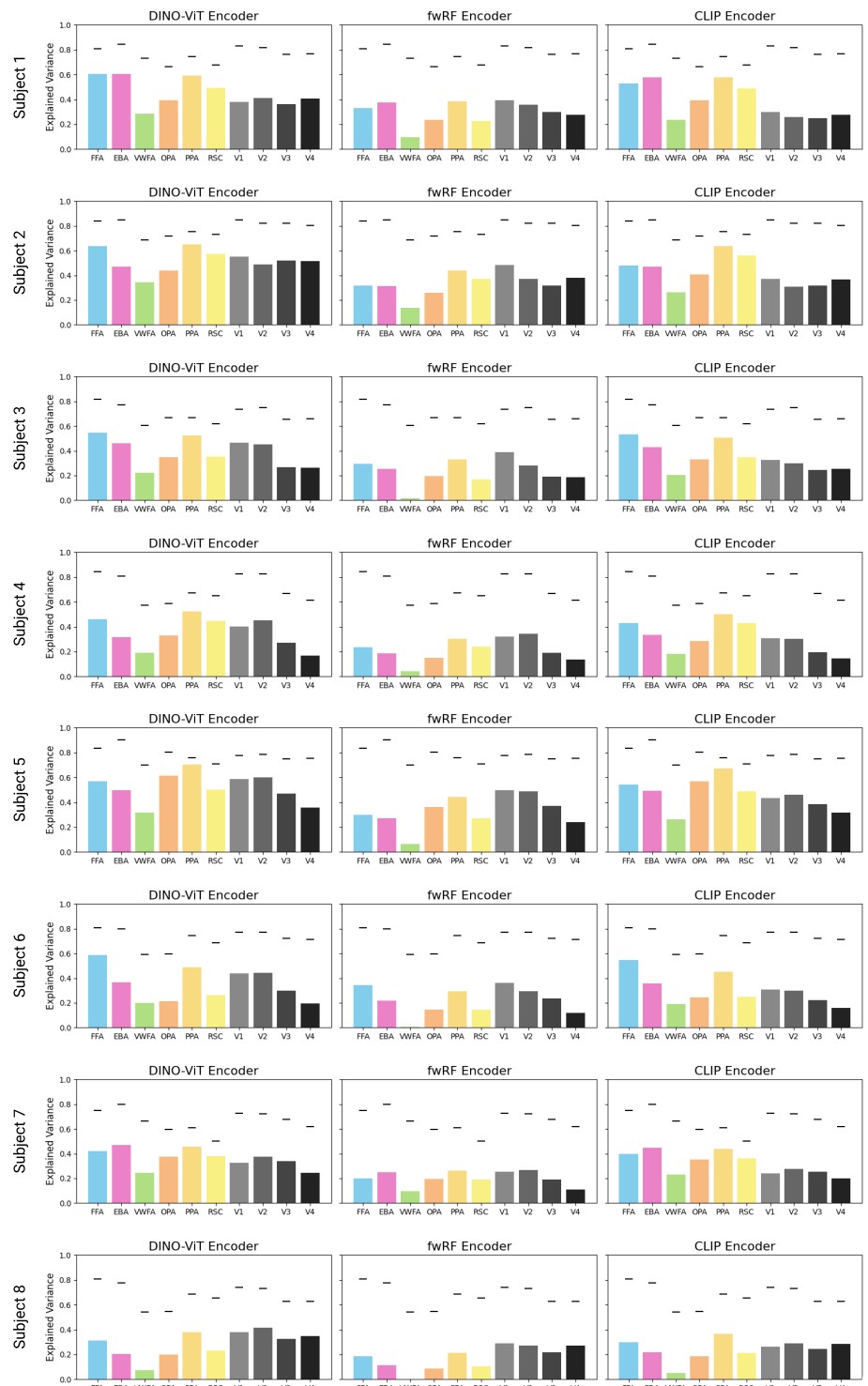

Figure 9: Encoding performance (explained variance) of DINO-ViT, fwRF, and CLIP encoders for each subject and each region of interest. Black lines indicate the estimated noise ceiling from Allen et al. (2022) (maximum over ROI voxels).

**DINO-ViT Encoder**. Adeli et al. (2023) explored the use of a pretrained 12-layer DINOv2 model (Oquab et al., 2023) as a feature extractor for a single-layer transformer that learns ROI-specific

queries, which are later linearly mapped to voxel activations. They train 22 models per subject, differing on the choice of layer used to extract features from DINOv2 and targeted ROIs. Finally, they employ an ensemble approach to produce their voxel-wise predictions. We simplify this process by fixing a single architecture and training it once for each category-selective ROI (9 models in total per subject). Specifically, we extract the output of each of the 12 layers in DINOv2 and pass each of them through a separate single-layer vision transformer (ViT) (Dosovitskiy et al., 2021). The output CLS token of each ViT is used by a multilayer perception (MLP) to predict voxel-wise activations for the ROI. The outputs of the 12 MLPs are aggregated through a learnable linear layer to produce our final predictions. All models are trained for 15 epochs with early stopping, using a learning rate of 1e-4 and a batch size of 64 samples.

**fwRF Encoder**. The Neural Encoding Dataset (NED) (Gifford and Cichy, 2024) provides pretrained brain encoders for the NSD dataset (Allen et al., 2022). These encoders are feature-weighted receptive field encoding models (St-Yves and Naselaris, 2018), neural networks trained end-to-end to predict neural responses.

## A.3 NSD Image Subsets

We identify six mutually exclusive subsets of images in NSD (Figure 10) to enforce diversity in our validation experiment. We filter the pixel-wise category annotations from COCO (Lin et al., 2014) as specified in Figure 11; each row also indicates the size of each subset for training and test sets.

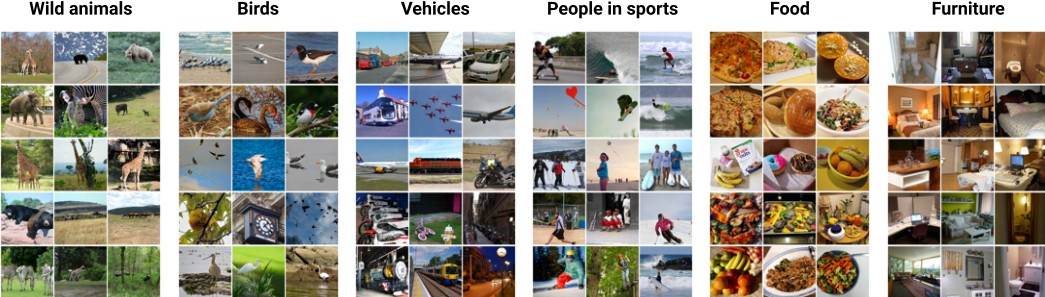

Figure 10: Example images per subset.

| Subset | Size (training set) | Size (test set) | COCO Categories |
|--------|---------------------|-----------------|-----------------|
| Wild animals | 773±38 | 150 | **include:** horse, sheep, cow, elephant, bear, zebra, giraffe
**exclude:** person, bicycle, car, motorcycle, airplane, bus, train, truck, boat, banana, apple, sandwich, orange, broccoli, carrot, hot dog, pizza, donut, cake |
| Birds | 125±14 | 24 | **include:** bird
**exclude:** person, bicycle, car, motorcycle, airplane, bus, train, truck, boat, banana, apple, sandwich, orange, broccoli, carrot, hot dog, pizza, donut, cake, horse, sheep, cow, elephant, bear, zebra, giraffe, cat, dog |
| Vehicles | 1000±37 | 123 | **include:** bicycle, car, motorcycle, airplane, bus, train, truck, boat
**exclude:** person, banana, apple, sandwich, orange, broccoli, carrot, hot dog, pizza, donut, cake, horse, sheep, cow, elephant, bear, zebra, giraffe, bird, cat, dog |
| People in sports | 839±31 | 101 | **include:** person AND frisbee, skis, snowboard, sports ball, kite, baseball bat, baseball glove, skateboard, surfboard, tennis racket
**exclude:** horse, sheep, cow, elephant, bear, zebra, giraffe, bird, cat, dog, bicycle, car, motorcycle, airplane, bus, train, truck, boat, banana, apple, sandwich, orange, broccoli, carrot, hot dog, pizza, donut, cake |
| Food | 509±33 | 52 | **include:** banana, apple, sandwich, orange, broccoli, carrot, hot dog, pizza, donut, cake
**exclude:** person, horse, sheep, cow, elephant, bear, zebra, giraffe, bird, cat, dog, bicycle, car, motorcycle, airplane, bus, train, truck, boat |
| Furniture | 883±40 | 108 | **include:** chair, couch, potted, bed, toilet
**exclude:** person, horse, sheep, cow, elephant, bear, zebra, giraffe, bird, cat, dog, bicycle, car, motorcycle, airplane, bus, train, truck, boat, banana, apple, sandwich, orange, broccoli, carrot, hot dog, pizza, donut, cake, dining table, umbrella |

Figure 11: Overview of the size and COCO categories used to define each subset. Upper rows (in the rightmost column) indicate categories present in all images. Bottom rows indicate categories that were explicitly excluded.

## A.4 ADDITIONAL IMAGE VARIATIONS

FUSIFORM FACE AREA (FFA)

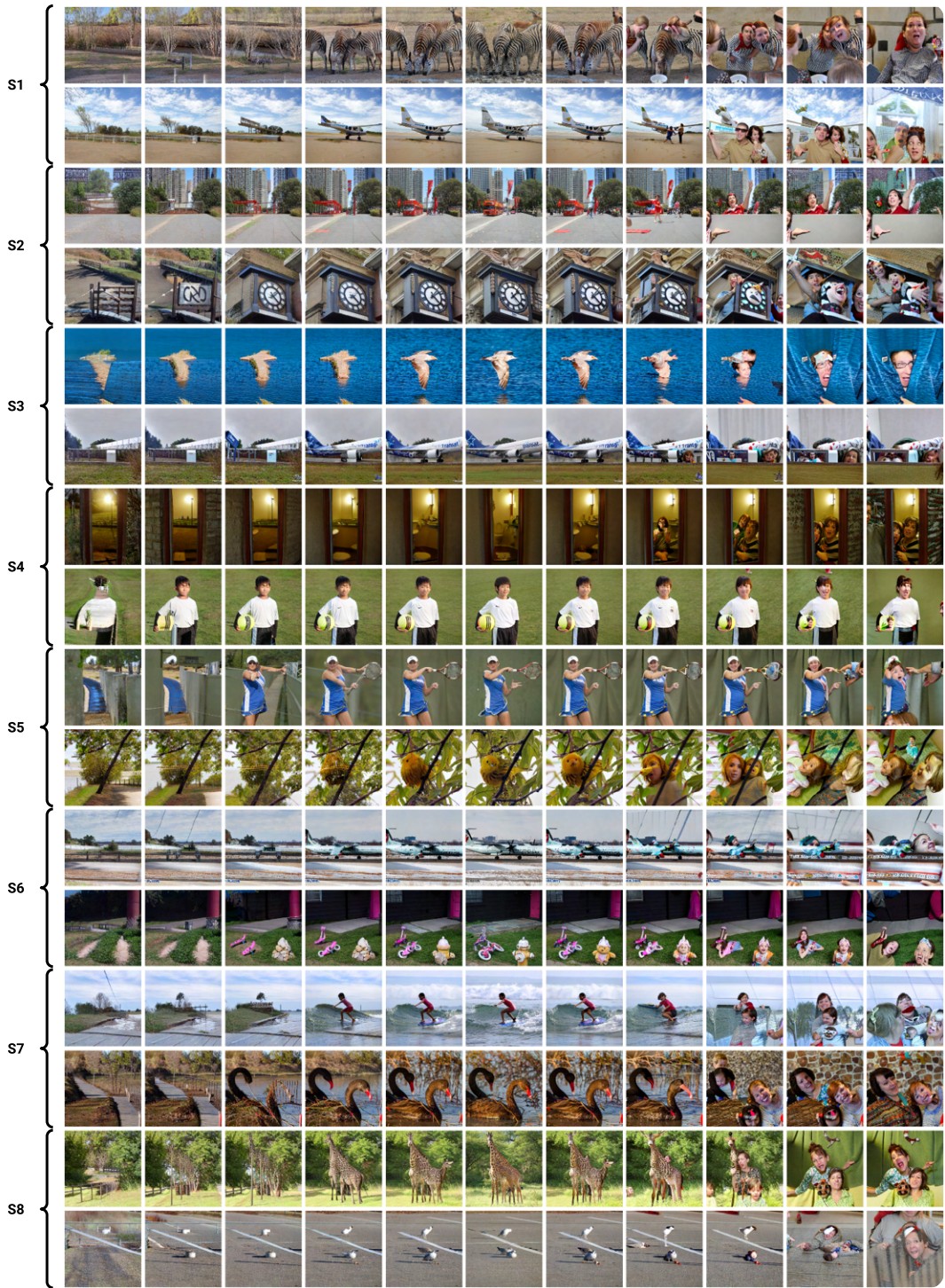

Figure 12: Example image variations (two per subject). Middle column: reference images. Left: minimization of FFA. Right: maximization of FFA.

EXTRASTRIATE BODY AREA (EBA)

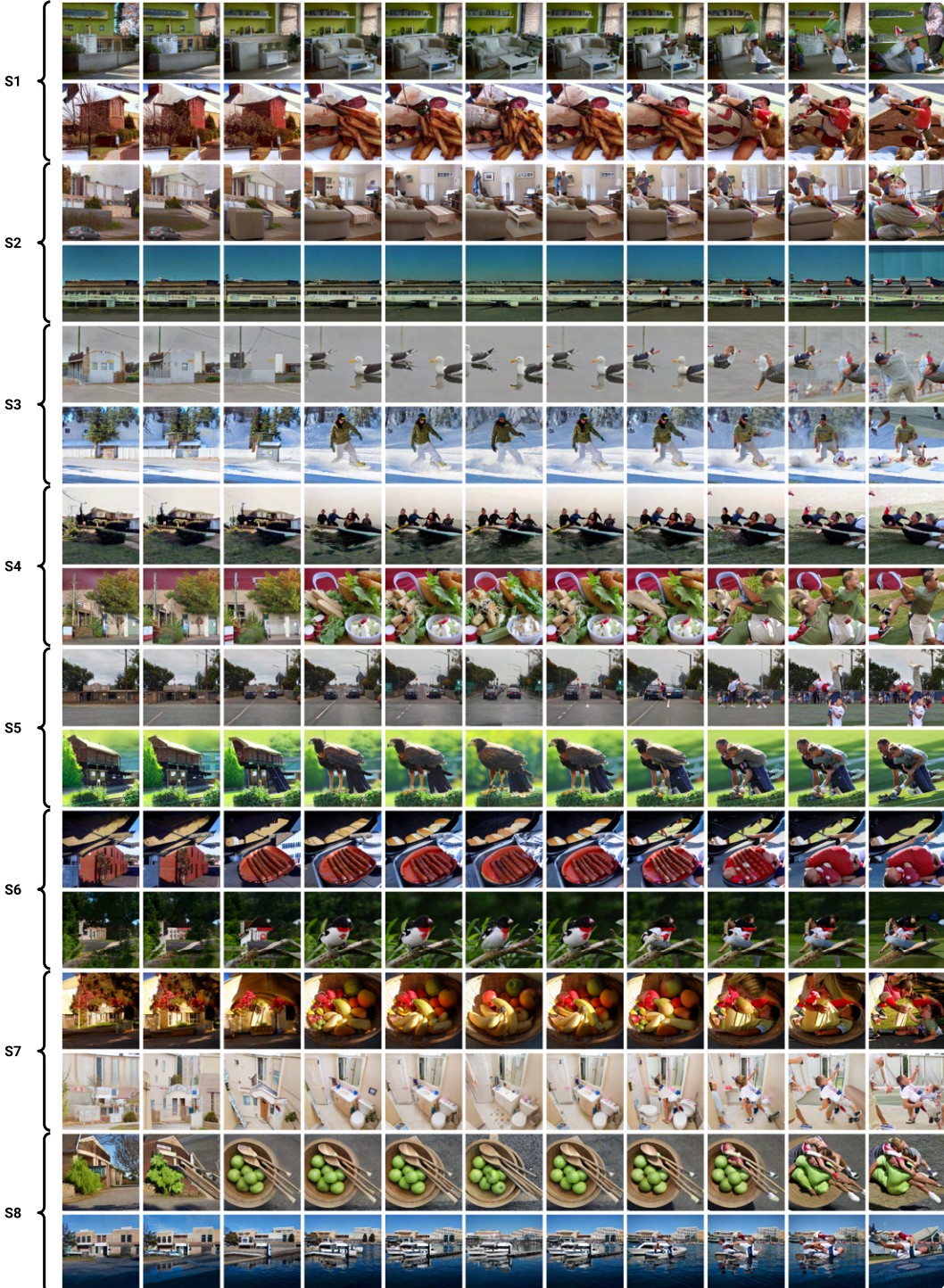

Figure 13: Example image variations (two per subject). Middle column: reference images. Left: minimization of EBA. Right: maximization of EBA.

VISUAL WORD FORM AREA (VWFA)

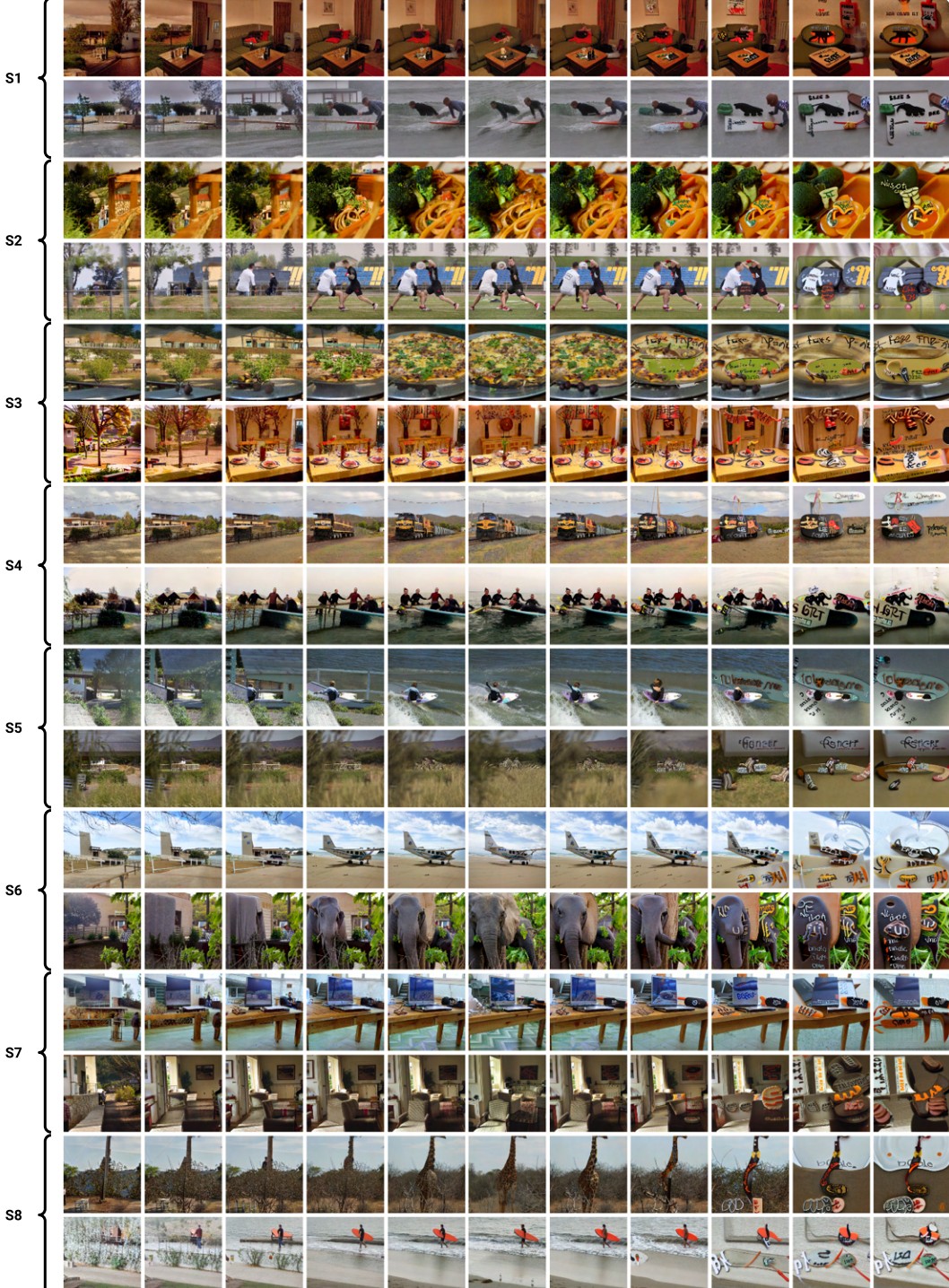

Figure 14: Example image variations (two per subject). Middle column: reference images. Left: minimization of VWFA. Right: maximization of VWFA.

OCCIPITAL PLACE AREA (OPA)

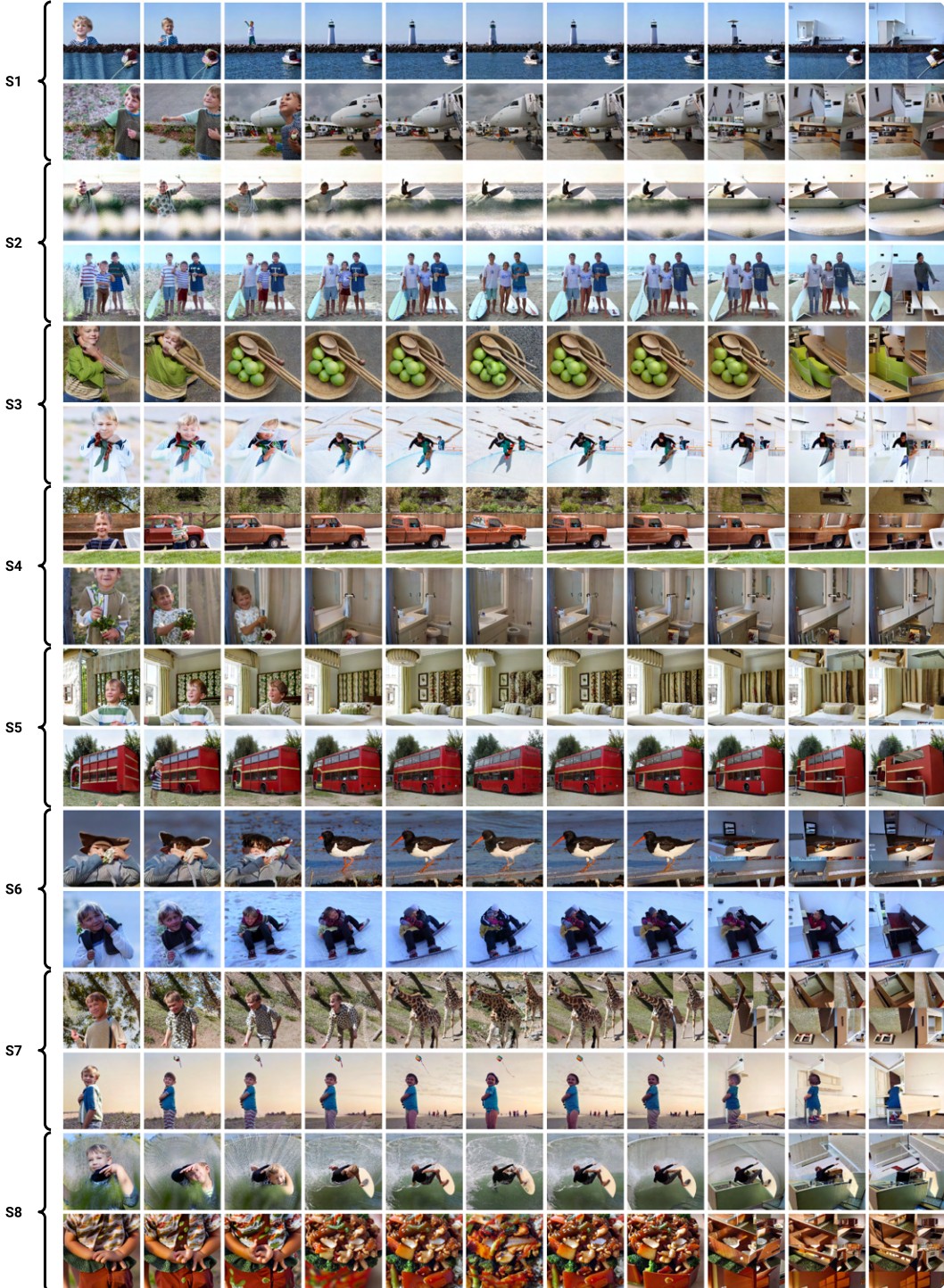

Figure 15: Example image variations (two per subject). Middle column: reference images. Left: minimization of OPA. Right: maximization of OPA.

PARAHIPPOCAMPAL PLACE AREA (PPA)

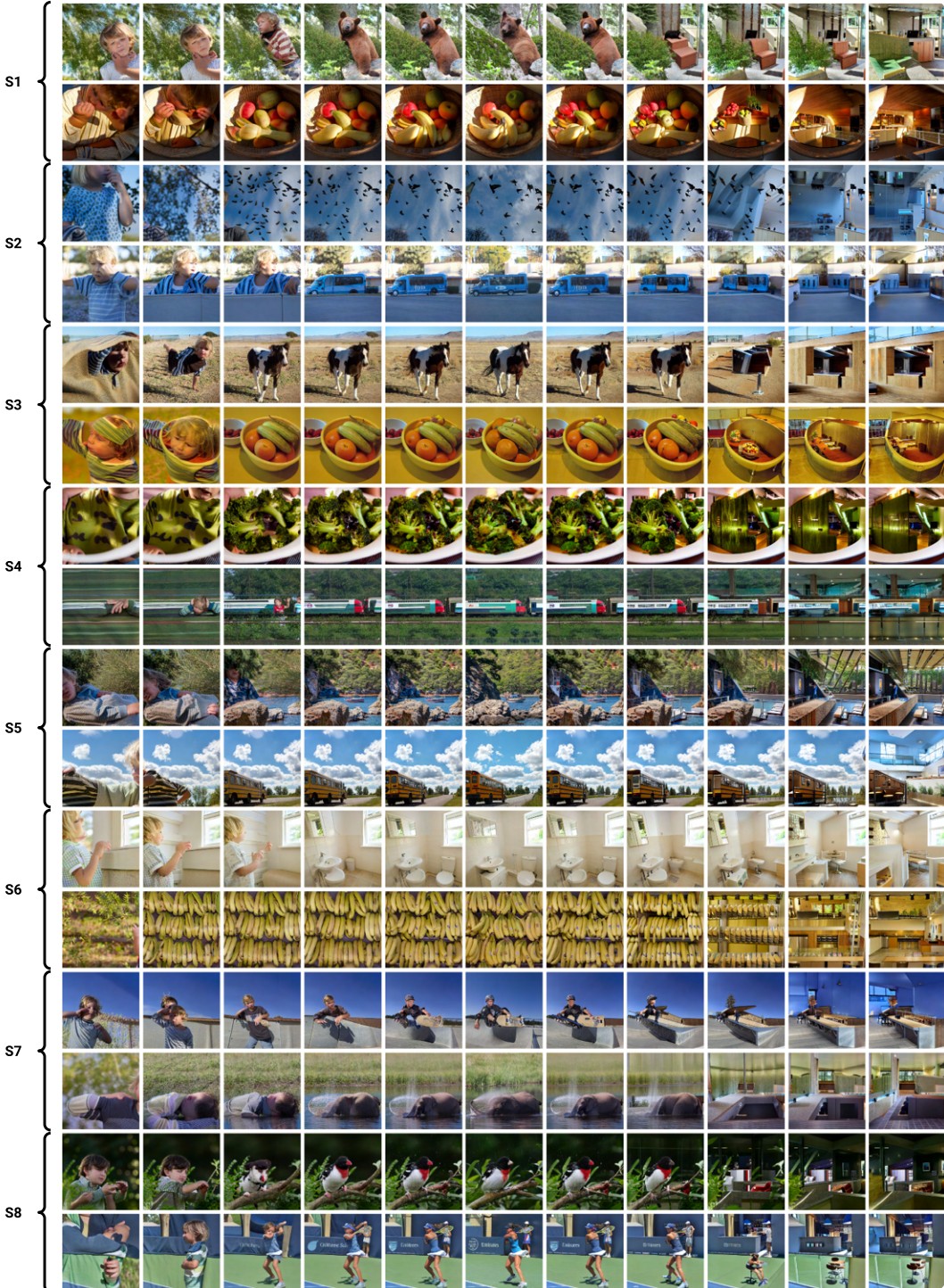

Figure 16: Example image variations (two per subject). Middle column: reference images. Left: minimization of PPA. Right: maximization of PPA.

RETROSPLENIAL CORTEX (RSC)

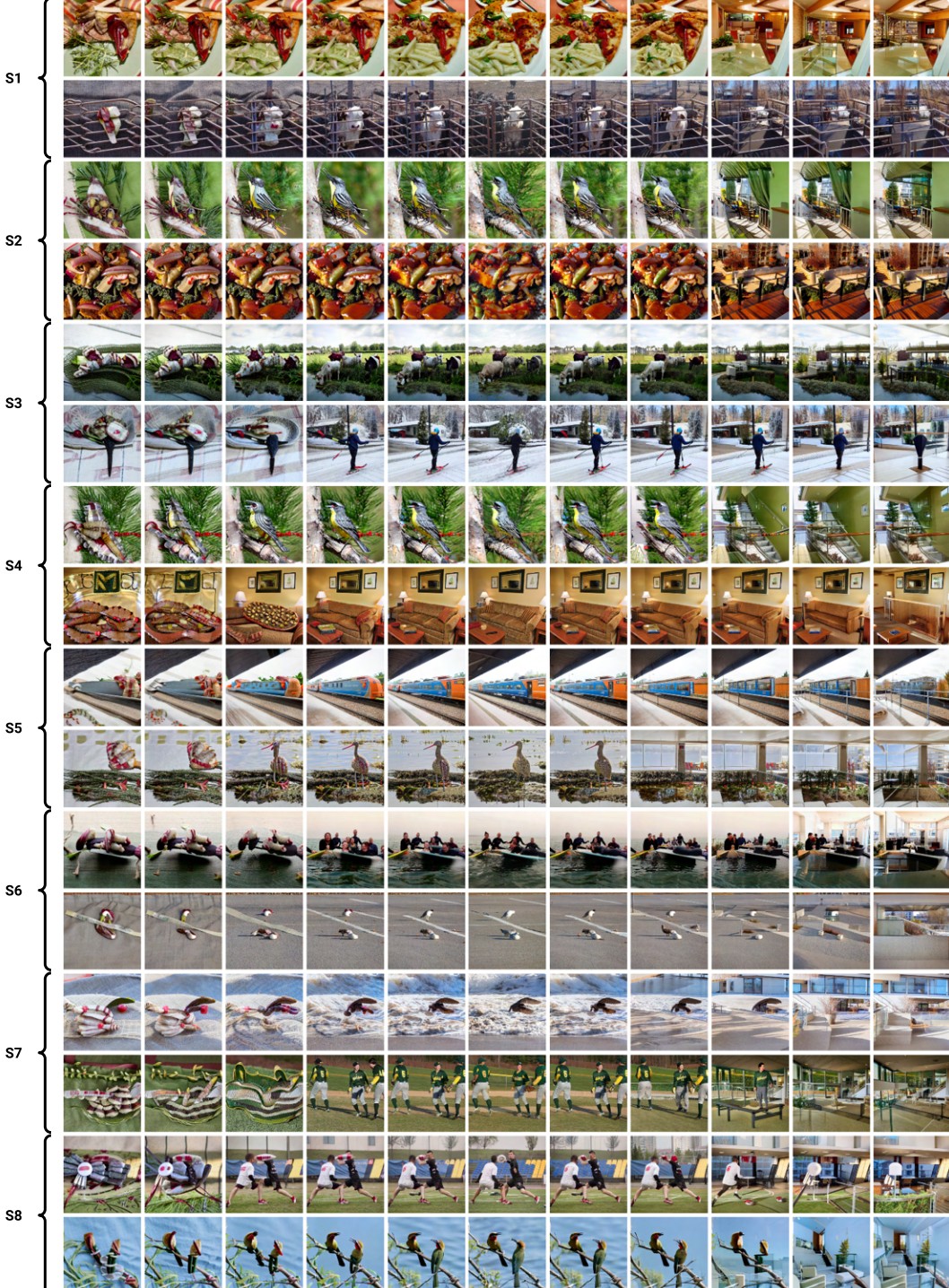

Figure 17: Example image variations (two per subject). Middle column: reference images. Left: minimization of RSC. Right: maximization of RSC.

## A.5 ADDITIONAL EXAMPLES: DIFFERENCES BETWEEN REGIONS

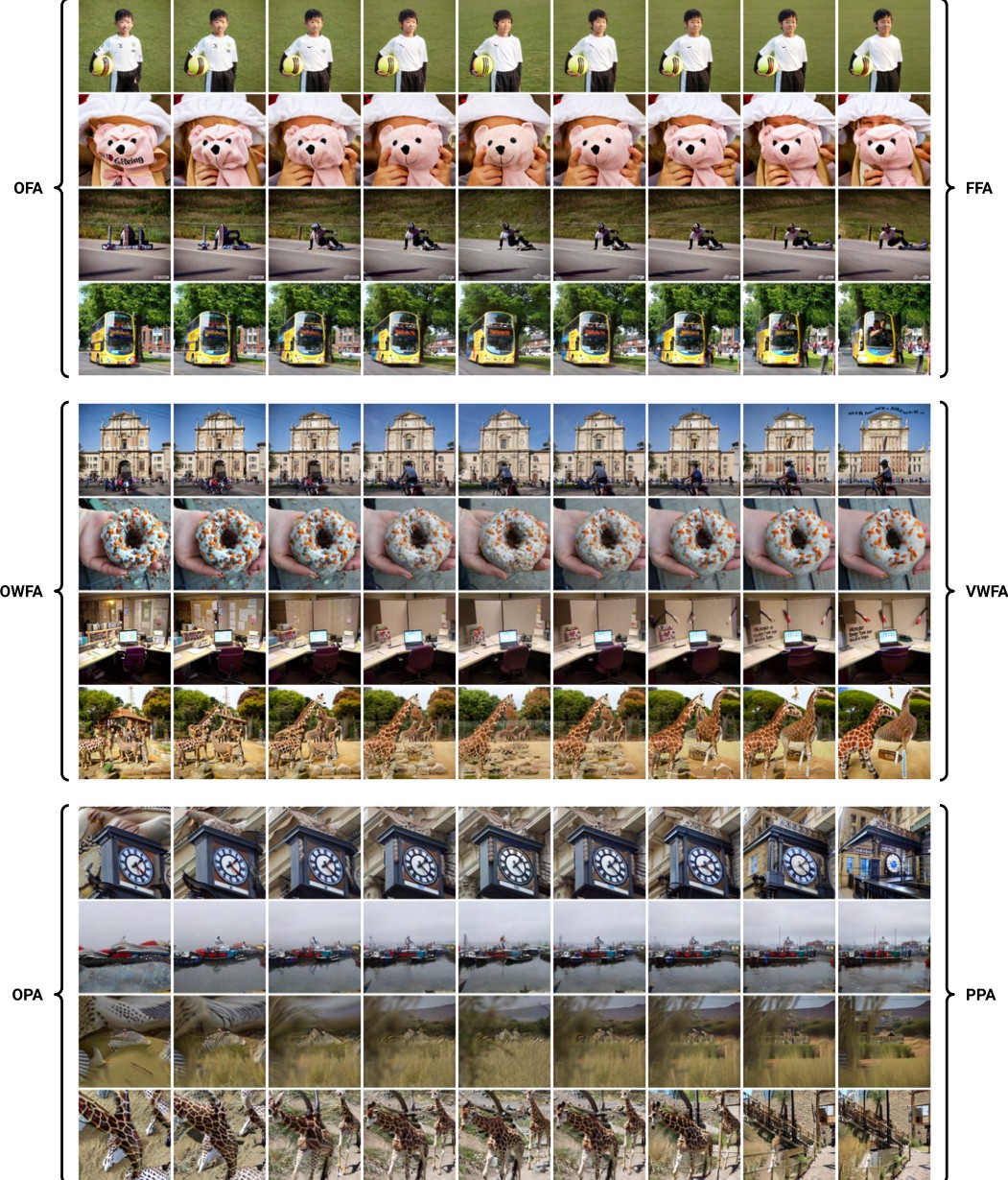

Figure 18: Example image variations accentuating one region (left) from another (right) in a reference image (middle).

## A.6  EARLY- AND MID-LEVEL ROIs

We showcase the use of BrainACTIV on early- to mid-level regions of interest. In particular, we target V1, V2, V3, and V4 for Subject 5. These regions are known to be selective for low-level image properties such as orientation (Tootell et al., 1998), ocular dominance (Menon et al., 1997), color (Engel et al., 1997), and spatial frequency (Mazer et al., 2002). We follow a similar procedure as outlined in subsection 4.2. Figure 21 shows example variations for each ROI, while Figure 19 shows successful modulation of predicted activations in these ROIs. Because BrainACTIV employs CLIP's image space to represent and modify the reference image's content, the manipulations mainly display semantic changes. However, these changes likely reflect semantic associations or co-occurrences with the low- and mid-level properties preferred by the ROIs, learned by CLIP during pre-training. For example, the appearance of light bulbs and cluttered elements in V1, or colorful objects in V4. Hence, conclusions and hypotheses formulated from these results must be cautious regarding semantic selectivities.

Figure 20 displays measured low- and mid-level image features for both optimal endpoints (averaged over all reference images). Importantly, we identify changes in color saturation and entropy (texture) that are not present for high-level ROIs (see subsection A.8).

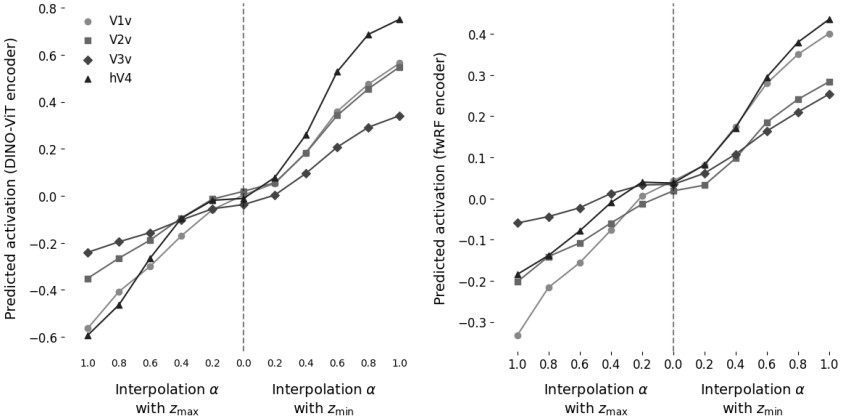

Figure 19: ROI activations predicted by DINO-ViT encoder (left) and fwRF encoder (right) as a function of interpolation $\alpha$ with each modulation embedding, averaged across test images.

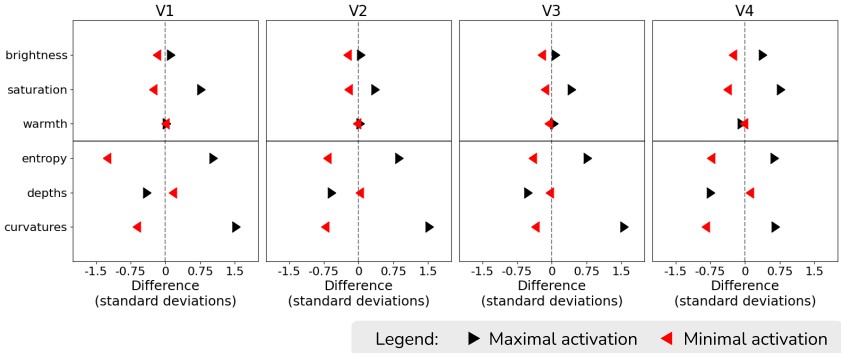

Figure 20: Quantification of average low-level and mid-level image features for variations at maximal (black) and minimal (red) activation.

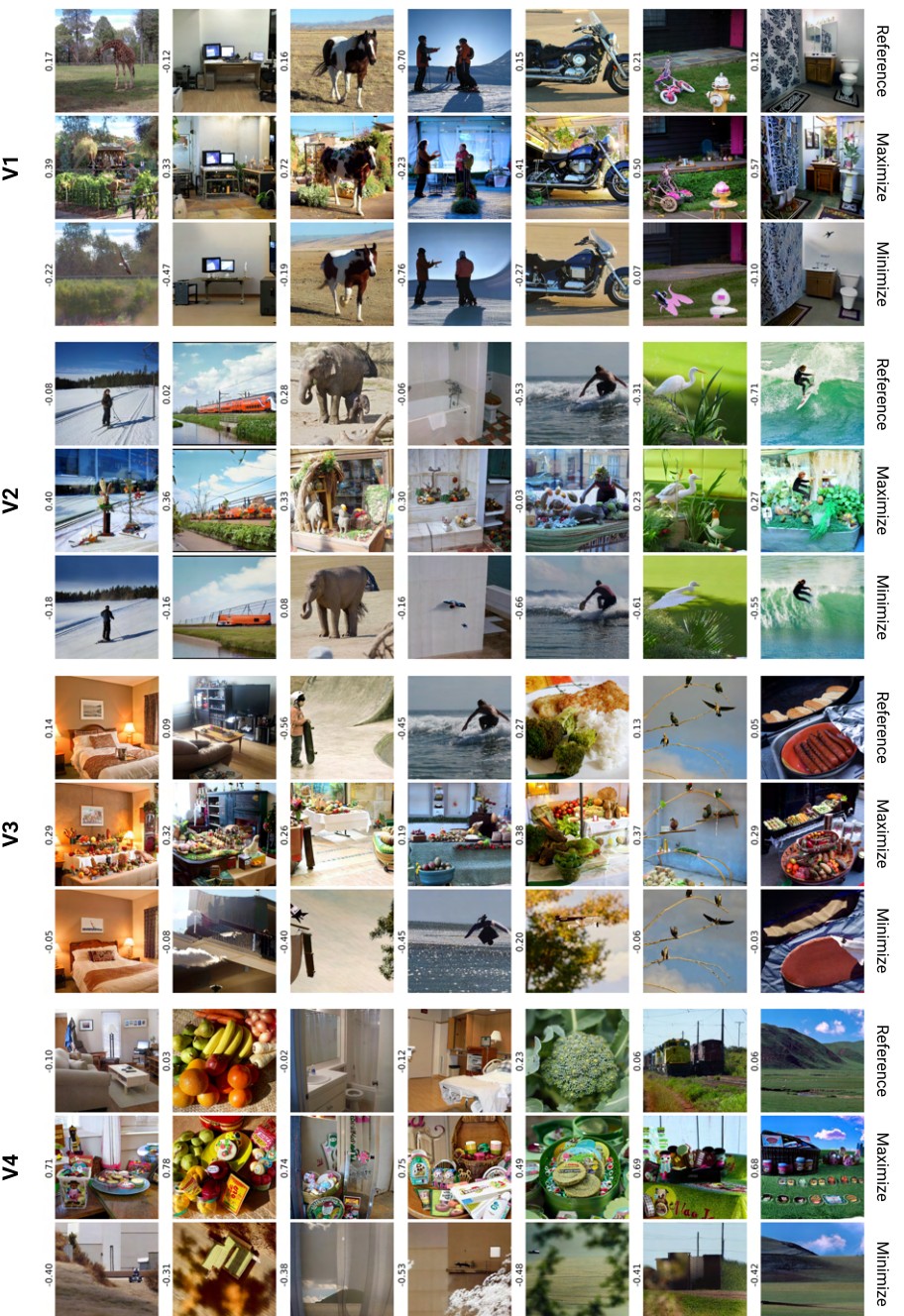

Figure 21: Example reference images (top) with their corresponding maximization (middle) and minimization (bottom) $\alpha = 1$ results. DINO-ViT predictions are displayed next to each image.

## A.7 EXPLORATION OF ANTERIOR IT CORTEX

We showcase the use of BrainACTIV to explore brain representations in regions for which selectivity is less well-understood than that of the areas targeted in subsection 4.2. Particularly, we target the anterior IT cortex. To do so, we first identify four components in the data by performing nonnegative matrix factorization (NMF) (Cichocki et al., 2009) on the NSD data matrix of anterior IT (number of images × number of voxels) for subject 5, similarly to Khosla et al. (2022)'s exploration of the ventral visual cortex. This operation yields two lower-dimensional matrices $W$ and $H$ whose product approximates the original data matrix. $W$ represents the relative contribution of each identified component to each of the voxels in anterior IT. $H$ represents the response of each component to all visual stimuli.

Figure 22 displays each component in $W$ overlayed on a flat cortical surface map. Higher values indicate a greater contribution of each component to the response profile of a voxel. Importantly, the four components define subdivisions of anterior IT that distinctly represent visual stimuli. Once we have identified the four components, we define sub-ROIs by taking the 100 most relevant voxels for each component and computing a modulation vector, as outlined in subsection 3.1. Then, we perform image manipulations as in subsection 4.2. Example results are displayed in Figure 24.

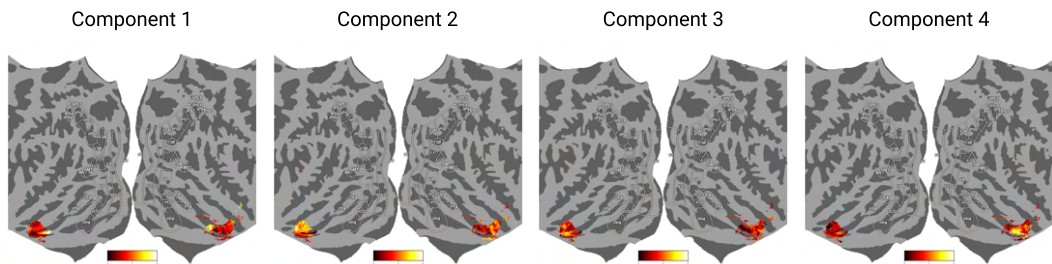

Figure 22: Relative contribution of each NMF component to the voxels in the anterior IT cortex.

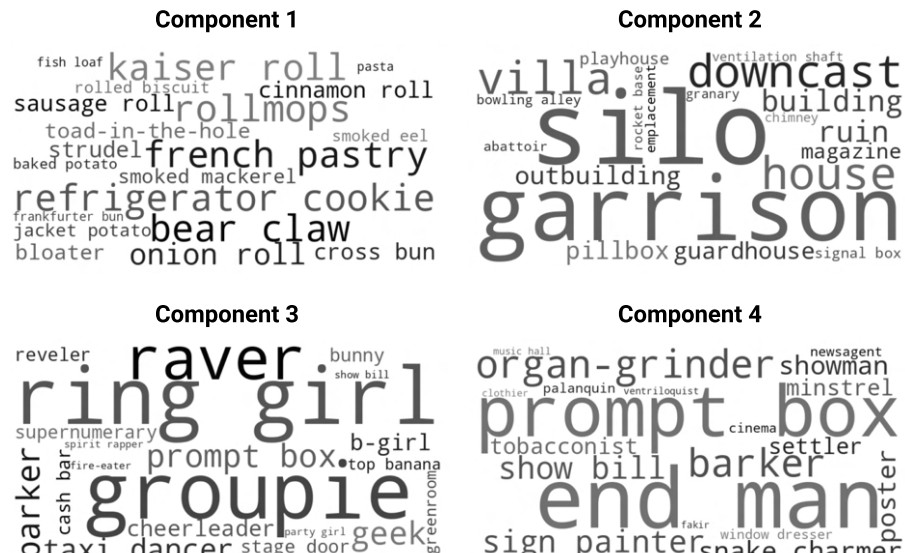

Figure 23: Top-nouns analysis of the image manipulations for each component illustrates what each of the corresponding sub-ROIs is suggested to be most responsive to.

To characterize the selectivity suggested by these results, we perform a top-nouns analysis in Figure 23. Inspection of these nouns together with manipulated images suggests the existence of subdivisions within anterior IT that are responsive to food, places, people, and text/objects. Interestingly, the visual characteristics of component 2 and 3 are different from what we observed for FFA, PPA, OPA, and RSC. For example, we seem to see gender-specific separation for people. These results can be used to formulate new hypotheses about neural representations in anterior IT, which must be tested through neuroimaging studies.

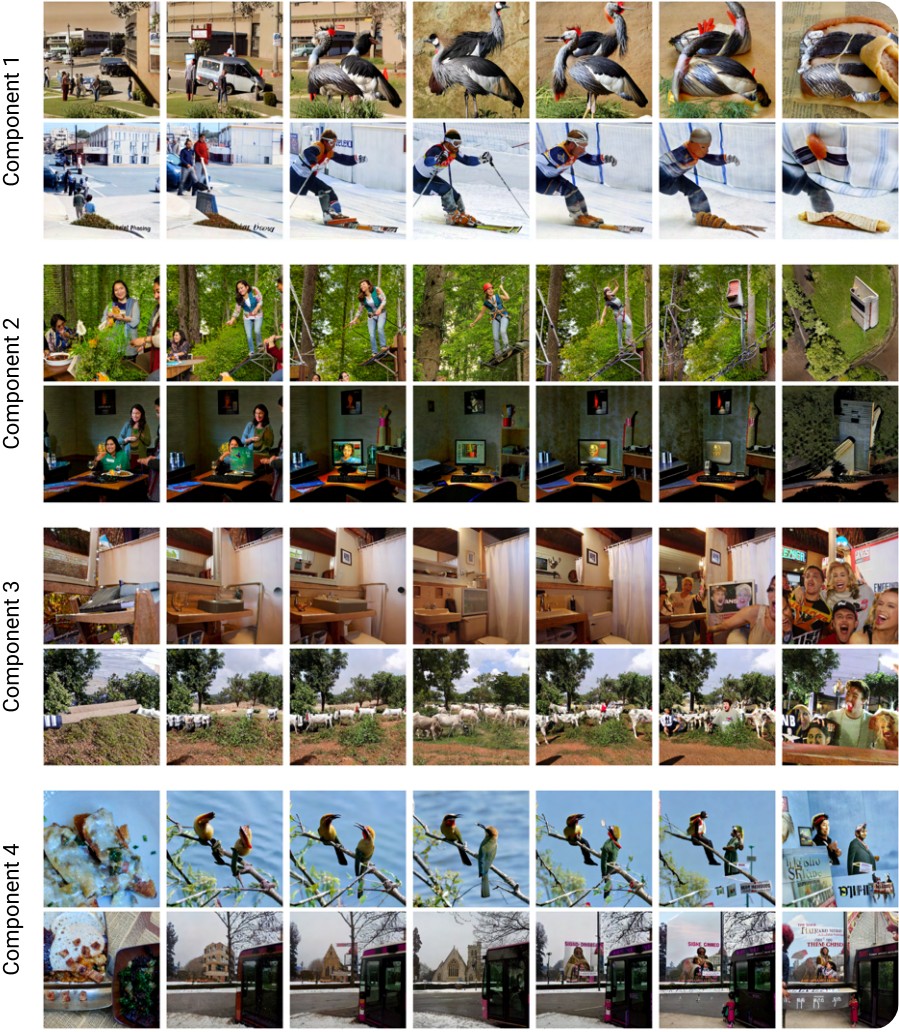

Figure 24: Example manipulations for each of the sub-ROIs computed for anterior IT. The middle column shows reference images; minimization results are on the left; and maximization results are on the right.

A.8    DIFFERENCE IN LOW-LEVEL IMAGE FEATURES

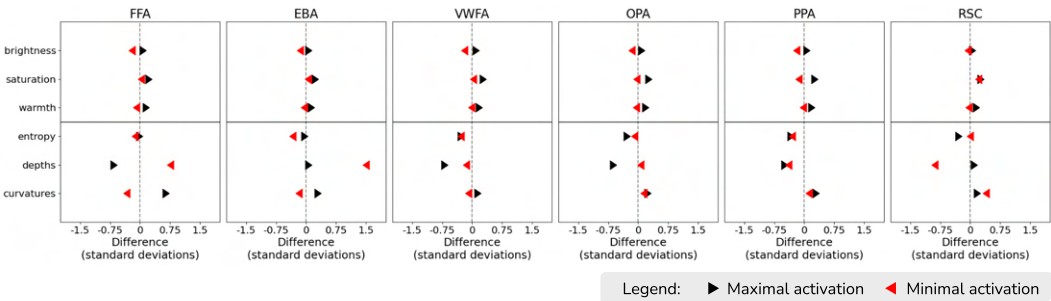

Figure 25: Quantification of low-level properties (brightness, saturation, color warmth) shows that these remain unaffected by variations at maximal (black) and minimal (red) activation, unlike mid-level features.

## A.9 SIMILARITY BETWEEN MODULATION EMBEDDINGS

BEFORE AVERAGING OVER SUBJECTS

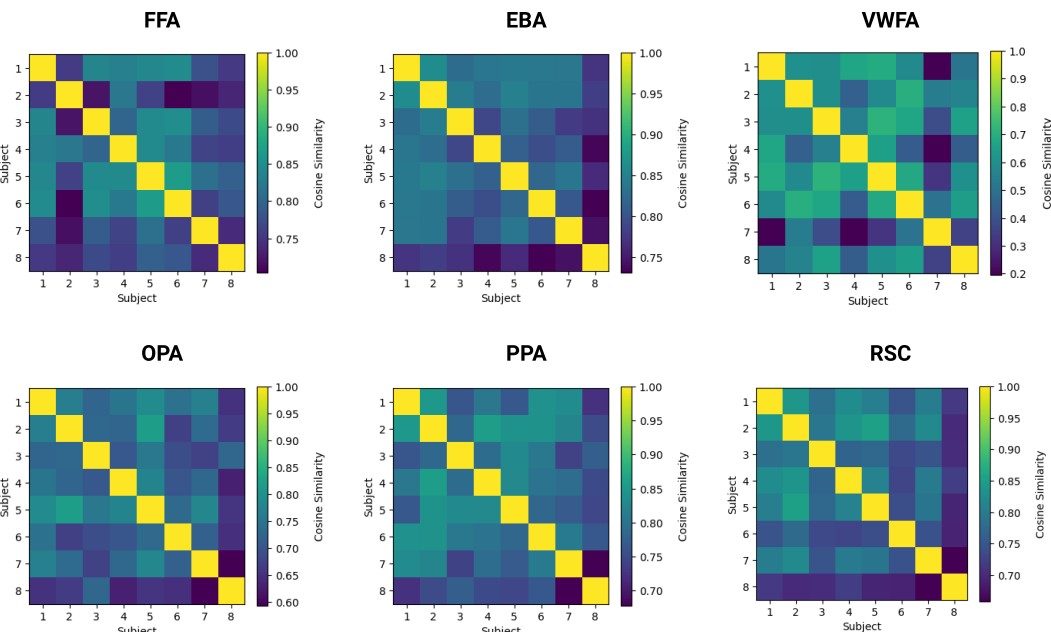

Figure 26: Cosine similarity between subject-specific modulation embeddings $z_{max}$ for each ROI before averaging over subjects.

AFTER AVERAGING OVER SUBJECTS

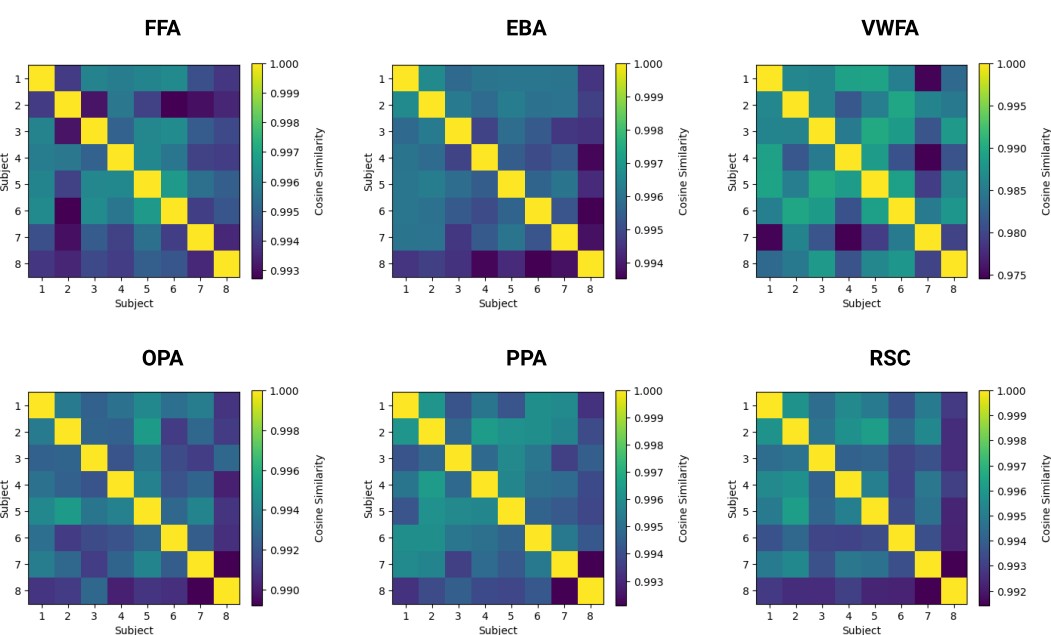

Figure 27: Cosine similarity between subject-specific modulation embeddings $z_{max}$ for each ROI after averaging over subjects.

### A.10 SIMILARITY ACROSS RANDOM SEEDS

FUSIFORM FACE AREA

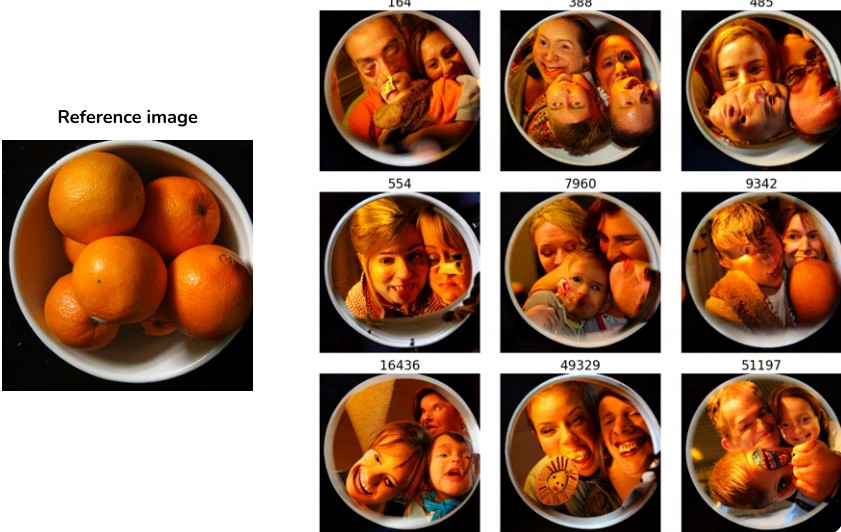

Figure 28: Example manipulations maximizing FFA ($\alpha = 1$) show great similarity across random seeds of the diffusion model, emphasizing that BrainACTIV isolates the effect of brain optimality.

PARAHIPPOCAMPAL PLACE AREA

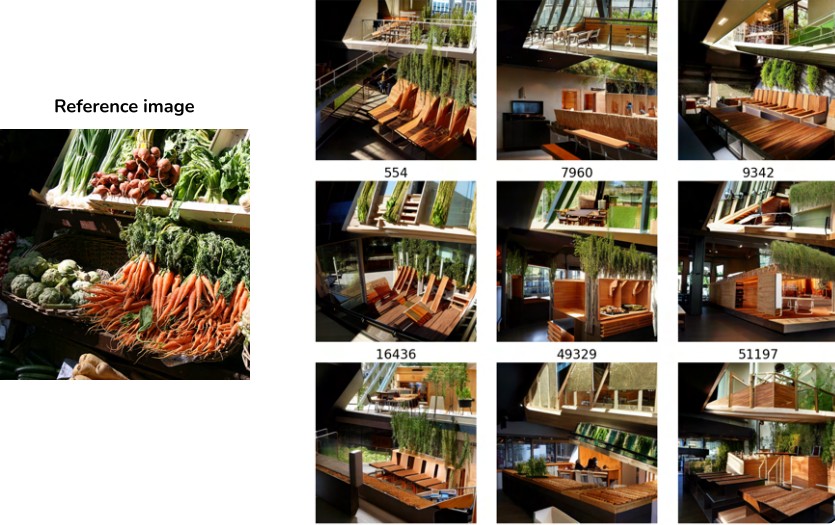

Figure 29: Example manipulations maximizing PPA ($\alpha = 1$) show great similarity across random seeds of the diffusion model, emphasizing that BrainACTIV isolates the effect of brain optimality.

### A.11    ILLUSTRATIVE NSD EXAMPLES PER MID-LEVEL FEATURE

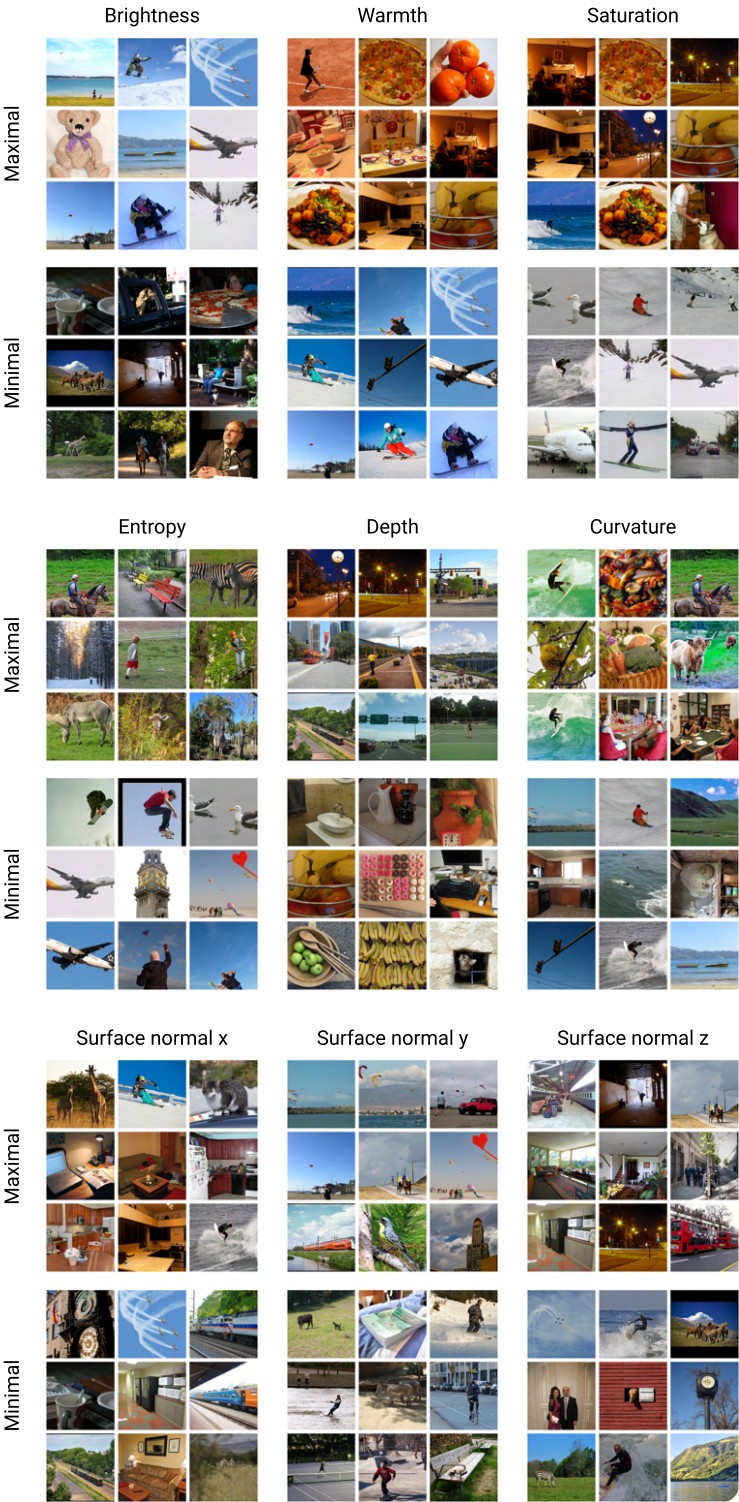

Figure 30: Examples from the NSD dataset displaying maximal and minimal pixel-averaged values for the different mid-level features we employ in this study.

### A.12 ADDITIONAL STRUCTURAL CONTROL BASELINES

To provide additional context on the structural control metrics for BrainACTIV (Table 1), we compute these metrics on stimuli synthesized without brain-conditioned targeting, namely $\alpha = 0$ (equivalent to simply passing the reference image through IP-Adapter, as would be commonly done for generating image variations).

We compute $L_2$ distance and LPIPS between the reference image and the synthesized images using different values of SDEdit's $\gamma$ (examples in Figure 31). Intuitively, the difference between these synthesized images is that $\gamma = 1$ shows how CLIP "interprets" the semantic content in the reference, while $\gamma = 0$ has no effect from CLIP and $\gamma$'s in-between are an interpolation between these two endpoints. Hence, all of these represent different interpretations of synthesis without brain conditioning.

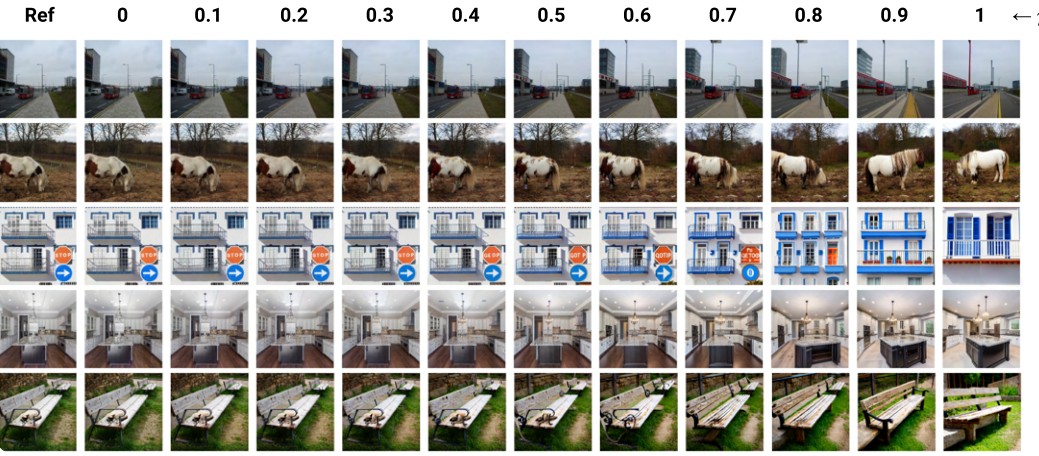

Figure 31: Examples for synthesis without brain conditioning using $\alpha = 0$ and differing $\gamma$.

Metrics in Figure 32 show that images closer to $\gamma = 0$ are structurally very similar to the reference image, while this similarity decreases as $\gamma$ approaches 1. In these plots, we highlight $\gamma = 0.6$, the maximal value used by BrainACTIV to compute results in Table 1.

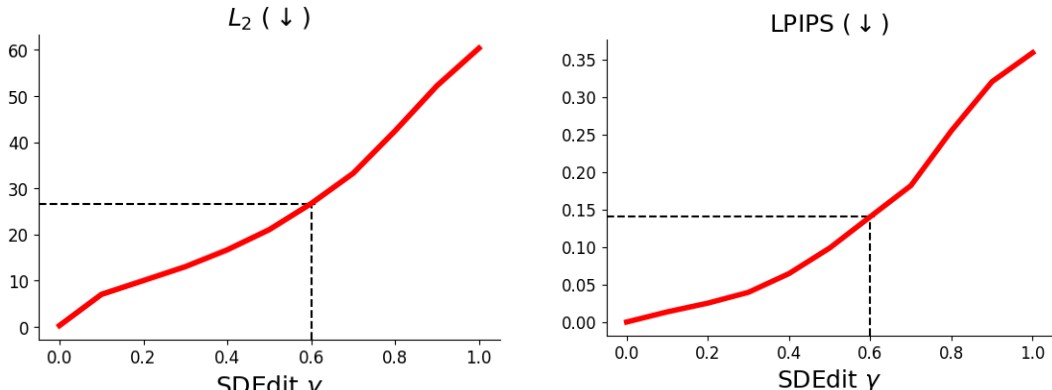

Figure 32: $L_2$ and LPIPS metrics for differing values of $\gamma$ when $\alpha = 0$. Averaged over test samples.

