# OpenReview forum: "BrainACTIV: Identifying visuo-semantic properties driving cortical selectivity using diffusion-based image manipulation"
_ICLR.cc/2025/Conference — ICLR 2025 Poster_

### Official Review · Reviewer_5W1K · 2024-10-25

**Soundness:** 3
**Presentation:** 3
**Contribution:** 3
**Rating:** 6
**Confidence:** 4

**Summary:**

The authors introduce BrainACTIV, a method for generating images modulated by the responses of a specific brain region, enabling interpolation between maximum and minimum activation levels. Additionally, they present two hyperparameters that regulate the semantic and structural variations of the generated images. The authors claim that this approach has the potential to provide insights into neuroscientific experiments.

**Strengths:**

1. The authors first achieved controllable image generation by manipulating a reference image to enhance or suppress specific target brain regions.
2. Based on the example images, particularly the interpolated manipulated reference images in Figure 1, the proposed method appears effective, especially concerning the functional regions of interest defined by neuroscience.
3. The article is highly readable, thanks to its clear writing and presentation.

**Weaknesses:**

1. The motivation needs to be better articulated. While the article emphasizes the controllable modification of the reference image, it lacks a clear rationale for this approach. What advantages does this controllable generation offer for guiding neuroscience discoveries compared to existing methods?
2. The technical novelty is limited, as the article primarily implements IP-Adapter and SDEdit to achieve controllable modifications of reference images.

**Questions:**

1. A comparison with related work [1-2] would be valuable if feasible.
2. The response to brain activation seems to be influenced by the fMRI encoder (specifically the CLIP encoder used in this paper), so its performance should be addressed in the experimental section.
3. How do the four mid-level features—entropy, metric depth, Gaussian curvature, and surface normals—contribute to advancements in neuroscience discoveries?
4. How do the two hyperparameters regulate the trade-off between semantics and structure? Increasing either hyperparameter seems to lead to deviations from the reference image, but it’s unclear how this establishes a trade-off between the two aspects. What is the rationale behind this approach?

[1] Gu Z, Jamison K W, Khosla M, et al. Neurogen: activation optimized image synthesis for discovery neuroscience[J]. NeuroImage, 2022, 247: 118812.

[2] Luo A, Henderson M, Wehbe L, et al. Brain diffusion for visual exploration: Cortical discovery using large scale generative models[J]. Advances in Neural Information Processing Systems, 2024, 36.

---

> ### Author Response · Authors · 2024-11-24
> **Response to Reviewer 5W1K (1/2)**
>
> We thank **Reviewer 5W1K** for the constructive review of our work. We address BrainACTIV’s motivation and concerns regarding technical novelty below. Further, we have carefully considered each of the comments and revised our paper accordingly.
>
> > ### **W1: Advantages of image manipulation**
>
> Our work indeed follows an existing line of research attempting to formulate new hypotheses about stimulus representation in the human visual cortex through data-driven analyses (facilitated by the recent availability of large naturalistic neuroimaging datasets). Existing methods generate images that maximize (predicted) activations of a target brain region, enabling the qualitative interpretation of visual/semantic properties that may be preferred by this region. However, none of them have explicitly enforced structural constraints on the generations; thus, these individual images have no explicit point of comparison. Consequently, their conclusions rely on the joint interpretation of a large number of images through, e.g., human behavioral studies to discern image features that are of relevance from those that were randomly generated by the model (hence irrelevant for driving activations).
>
> The introduction and modification of a reference image provides a direct comparison point for each synthesized stimulus, bringing about several benefits:
>
> - Straightforward interpretation of image features that are relevant to the brain region (because random variability from the diffusion model is minimized).
> - Possibility to quantify these relevant features through automated methods from computer vision (because the values for reference and variation images are directly comparable). Hence, there is no longer a need for human subjects to interpret hundreds of images, and the computational costs are vastly reduced.
> - The produced image variations represent a hypothesized tuning axis for the target region or group of voxels, with the reference image serving as a control stimulus. Hence, BrainACTIV can be employed by researchers in novel neuroscientific studies to study fine-grained selectivity properties.
>
> To make this motivation clearer in the text, **we have rephrased several sentences in the Abstract and Introduction, and we have reordered and expanded previous Results sections 4.3 to 4.5 to better convey these advantages**.
>
> > ### **W2: Technical novelty**
>
> We acknowledge that BrainACTIV primarily implements existing methods for controllable image generation. However, we believe in the importance of bringing tools and ideas from machine learning closer to adjacent scientific fields (particularly, computational neuroscience), which is why we marked ‘applications to neuroscience/cognitive science’ as our primary area The fast-paced development of new models and methods in machine learning leaves many potential applications unexplored, missing out on significant impacts to these fields. We argue that BrainACTIV represents an innovative application of controllable diffusion-based image generation to advance our understanding of the human visual cortex.
>
> BrainACTIV is the first work to explore image variation with respect to a reference image using generative models to (1) formulate novel hypotheses about stimulus representation in the human visual cortex and (2) produce controlled experimental stimuli for novel neuroscientific experiments.
>
> In addition, the incorporation of IP-Adapter and SDEdit significantly reduces the computational costs needed to generate conclusions, relative to BrainDiVE. Our image generation process takes ~80 hours for all subjects on an NVIDIA A100, while Luo et al. (2023) [b] report 1,500 hours on an NVIDIA V100 (reports indicate A100 is only ~2 to 4x faster than V100 [a], while our results take ~9x less time, accounting for the difference in Stable Diffusion versions). **We have briefly mentioned this advantage in line 350**. Moreover, we remove the need for human behavioral studies to analyze the generated images, further saving time and costs.
>
> ---
>
> - [a] https://lambdalabs.com/blog/nvidia-a100-vs-v100-benchmarks
> - [b] Luo, A. F., Henderson, M. M., Wehbe, L., and Tarr, M. J. (2023). Brain diffusion for visual exploration: Cortical discovery using large scale generative models. NeurIPS.

---

> > ### Author Response · Authors · 2024-11-24
> > **Response to Reviewer 5W1K (2/2)**
> >
> > > ### **Q1: Comparison to BrainDiVE and NeuroGen**
> >
> > While BrainACTIV presents methodological improvements upon BrainDiVE [b] and NeuroGen [c], the lack of a reference image in these works precludes a direct comparison. Moreover, a comparison in terms of image generation quality is not intended, since the three methods can be straightforwardly adapted to any state-of-the-art generative model (diffusion models for BrainACTIV and BrainDiVE, GANs for NeuroGen). Because BrainACTIV retains the strengths presented by these approaches (e.g., fine-grained distinction of similar ROIs, as presented in Section 4.4), we argue that the superiority of BrainACTIV lies in the possibilities opened up by our image manipulation approach, as outlined above. However, we are open to suggestions regarding analyses that could provide a fair comparison between these methods.
> >
> > > ### **Q2: Performance of CLIP encoder**
> >
> > We strongly agree that the performance of the CLIP encoder provides important context for our results. **Therefore, we have adapted Figure 9 to display the performance of the three brain encoders on the held-out test set for all subjects and ROIs. We have also added the corresponding CLIP encoder predictions to examples in Figure 3(A), as well as predicted modulation results from the CLIP encoder in Figure 3(B)**.
> >
> > > ### **Q3: Mid-level features**
> >
> > These four mid-level features are not intended to provide a comprehensive analysis of the role of mid-level features representation in the human visual cortex, but rather to serve as an illustration of how BrainACTIV, because of its use of a reference image as baseline, can give precise quantitative metrics of the effects of brain optimization on images and on predicted brain activations. We have **updated Methods section 3.2 to better explain our intention here**, while we have also **supplemented Results section 4.3 with additional references** to relate our observed effects of mid-level features on predicted brain activations to existing neuroscience findings.
> >
> > > ### **Q4: Trade-off between semantics and structure**
> >
> > Thank you for raising these questions. **We have revised the original explanation of the two hyperparameters in our text (now Section 4.5) to explain their role more clearly, and we have rephrased the term “trade-off” to avoid confusion**. Moreover, **this section now emphasizes the use of BrainACTIV to produce novel experimental stimuli**.
> >
> > To clarify these points: both hyperparameters indeed introduce deviations from the reference image, decreasing semantic (CLIP) and structural similarity (LPIPS) to the latter. However, when we look at the two different endpoints (one hyperparameter higher than the other) we are presented with two choices: (1) retain structural similarity while varying semantics or (2) having more freedom in the low-level structure while representing very similar semantic content. Both of them present opportunities when testing neural representations in controlled experiments (e.g., to study the relative contribution of visual versus semantic features, as mentioned in our Introduction). Hence, the rationale behind this section is to briefly present both options to researchers and illustrate their effect on the image.
> >
> > ---
> >
> > - [b] Luo, A. F., Henderson, M. M., Wehbe, L., and Tarr, M. J. (2023). Brain diffusion for visual exploration: Cortical discovery using large scale generative models. NeurIPS.
> > - [c] Gu, Z. et al. (2022). NeuroGen: Activation optimized image synthesis for discovery neuroscience. NeuroImage, 247:118812.

---

> > > ### Comment · Reviewer_5W1K · 2024-11-26
> > >
> > > Thank you to the authors for the detailed clarifications, which addressed most of my questions. However, I remain confused about the motivation behind the reference image.
> > >
> > > From my understanding, the reference image serves two purposes: 1. controllable image generation and 2. tuning mid-level features. As the authors mentioned, mid-level features help explain the reference image as a baseline. However, I am still uncertain about the necessity of controllable generation.
> > >
> > > I completely agree with the authors regarding the random variation in the diffusion model. However, since the results are fixed under the same random seed, horizontal comparisons of the results are possible. In this context, the reference image is determined by the random seed. While this paper extends the customization of the reference image, I still do not fully understand the advantages of doing so.

---

> > > > ### Author Response · Authors · 2024-11-28
> > > >
> > > > Thank you for the opportunity to clarify the motivation for the reference image. In short,
> > > >
> > > > - **(1)** The reference image is a technical requirement for interpolating in CLIP space (Equation 5) to provide the necessary high- and mid-level information to the synthesized image variations through IP-Adapter.
> > > >
> > > > - **(2)** We control the low-level structure of the synthesized image variations by using SDEdit on a *real* reference image (as opposed to only fixing the random seed) for better integration of BrainACTIV into neuroscientists’ own experiments.
> > > >
> > > > For clarification: in all our analyses, we use (real) test images from the Natural Scenes Dataset as reference inputs.
> > > >
> > > > **Our latest revision incorporates slight modifications to lines 74, 79, and Sections 3.1, 4.2, and 4.5 with the aim of improving the presentation of the above information**. We provide a more elaborate explanation for this below.
> > > >
> > > > BrainACTIV's image generation process depends on two components:
> > > >
> > > > 1. IP-Adapter for providing the (brain-optimized) *high-level* and *associated mid-level* information to the synthesized image variations.
> > > >
> > > > The adapter inputs our intermediate embeddings (Equation 5) as conditions to the diffusion model. These embeddings are obtained through spherical interpolation in CLIP space between a visual stimulus z_I and a brain-derived optimal endpoint z_max (from Equation 2). As such, we require a reference visual stimulus to perform this computation. Moreover, as you mention, this reference indeed acts as a baseline for the synthesized stimuli in terms of high and mid-level information.
> > > >
> > > > 2. SDEdit to obtain and fix the initial diffusion latents, thus reproducing the *low-level* (structural) information of the reference image in the synthesized image variations.
> > > >
> > > > You are correct in pointing out that fixing the diffusion model's latents through a random seed would already impose the same low-level structure to all image variations, directly allowing comparisons (to the reference) in terms of pixel-wise mid-level features and semantic content (assuming the reference image would also be generated by the diffusion model). In that setting, one could indeed refrain from using SDEdit and obtain similar results for our analyses (i.e., Figures 4 and 5).
> > > > However, we opt to use SDEdit on a real reference image—as opposed to fixing the latents through a random seed—for better integration of BrainACTIV into neuroscientific hypothesis-driven experiments, enabling researchers to input any pre-selected control stimulus.
> > > >
> > > > > ### Concrete example
> > > > > These experiments could test hypotheses about semantic tuning that are minimally confounded by low-level structural properties. To give a concrete example: in neuroscientific research on face perception, it continues to be debated whether brain responses to images containing faces reflect tuning to ‘face-specific’ features or rather to domain-general object features (e.g. curved contours); see Vinken et al., (2023) ([a], also cited in Introduction). In this setting, BrainACTIV's incorporation of a user-input reference image a) allows it to serve as a control stimulus and brain activation baseline to which newly measured brain responses to generated stimuli can be compared, and b) to keep constant the presence of domain-general features while only changing face-specific features, thus allowing for the potential isolation of the effect of such features on brain responses in face-selective brain regions.
> > > >
> > > > We hope this information has clarified the motivation behind the reference image, and we welcome further conversation regarding it.
> > > >
> > > > ---
> > > >
> > > > - [a] Vinken K., Prince JS,  Konkle T, and Livingstone MS. The neural code for “face cells” is not face-specific. Sci. Adv. 9 ,eadg1736 (2023). DOI:10.1126/sciadv.adg1736

---

> > > > > ### Comment · Reviewer_5W1K · 2024-11-28
> > > > >
> > > > > Thanks for the clarification. The face perception case study illustrates the motivation of BrainACTIV well, which is to manipulate "real images" with brain patterns to study "specific” neural representations. The authors propose spherical interpolation in CLIP space to provide conditional embedding for IP-adapter to achieve the above goal.
> > > > >
> > > > > I would increase the rating score due to its solid approach and insights into neuroscience research.

---

### Official Review · Reviewer_eXwK · 2024-10-31

**Soundness:** 4
**Presentation:** 4
**Contribution:** 4
**Rating:** 8
**Confidence:** 4

**Summary:**

The paper proposes a method for manipulating a reference image so as to selectively enhance (or decrease) the corresponding predicted brain activity in a given target region. The image manipulation is performed with pretrained diffusion models, and the brain activity prediction relies on training a brain decoder to perform regression between a large image dataset (NSD) and the corresponding brain activation patterns.

**Strengths:**

* The proposed image manipulation technique is non-trivial, and goes much beyond deploying an off-the-shelf diffusion system. It leverages state-of-the-art diffusion systems and adapters.
* The manipulated images are convincing, and compatible with known selectivity in the target brain regions.
* The technique allows to draw further subtle distinctions between brain regions that have been traditionally associated with similar category selectivity, and could be used to inform future neuroscience experiments.
* The method allows to control the trade-off between semantic variation and structural similarity with two hyperparameters.

**Weaknesses:**

The paper shares weaknesses with most brain decoding studies, in the sense that the results describe predictions of a model trained from a specific dataset and a specific (combination of) deep learning model(s).  These predictions will ultimately require validation from actual experiments. However, this is well acknowledged in a paragraph on “limitations”.

**Questions:**

* “we opt to average the embeddings over all subjects, excluding the subject on which predictions are made. Hence, we modulate brain activity in each subject through a signal derived exclusively from the rest of the subjects’ data”: This statement is vague enough to be interpreted in different ways (as there are multiple components in the pipeline that could be calculated over one or multiple subjects). I would suggest clarifying things with the corresponding variable names from equations (1-5).
* On line 224, the methods description switches from computation of variables related to the target region of interest (e.g. $z_{max}$), to the introduction of the specific reference image and its manipulations $z_I$. It would help to state this explicitly.
* Table 1 provides structural metrics for manipulated images together with a lower baseline computed from random images. I would suggest adding an upper baseline calculated from diffusion-based image variations without brain signal conditioning.
* In Figure 6, it could be helpful to include the reference image (e.g. in the top-left corner).
* The literature review on “optimal visual stimulus generation” misses (at least) one reference from the BrainDiffuser paper (Ozcelik et al, 2023).

---

> ### Author Response · Authors · 2024-11-24
> **Response to Reviewer eXwK**
>
> We appreciate the positive feedback and helpful suggestions from **Reviewer eXwK**. We address each of them below.
>
> > ### **Q1 and Q2: Clarity in text**
>
> Thank you for this valuable feedback. **We have updated lines 228 and 345 to make these variables more explicit in the text**.
>
> > ### **Q3: Additional baseline in Table 1**
>
> We agree that the baseline of random images constitutes a lower baseline and that it would be useful to have an upper baseline as well, to quantify the penalty on structural control associated with brain-conditioning . However, we are not entirely sure what insight we would get from comparing to variations without brain conditioning. In our view, this would emphasize that the structural control as implemented by SDEdit is working correctly, but the distance values would be really close to zero since the variation and the target would be almost identical. We welcome further elaboration on the added value of this suggested baseline or other potentially useful baselines.
>
> > ### **Q4: Improvement to Figure 6 (now Figure 7)**
>
> We agree in adding the original image in old Figure 6 (now Figure 7). We have omitted the row and column corresponding to $\alpha=0$ and $\gamma=0$ from the examples, since these are highly redundant. **We have added the reference image in Figure 7(A) within a simple graphic explaining the usefulness of BrainACTIV for producing novel neuroscientific stimuli**. This accompanies changes made to Section 4.5 to emphasize this contribution based on reviews.
>
> > ### **Q5: Literature review missing references**
>
> Thank you for pointing this out. We acknowledge the vast amount of work performed on image reconstruction from brain activity, and its connection to our work as an alternative way to perform brain-guided image synthesis. **Accordingly, we have added relevant citations in our Related Work section**. Particularly, the work by Ozcelik et al. (2023) [a] is a good example of image reconstruction methods that can be adapted to generate optimal inputs for specific ROIs. **We have added citations to this work in lines 60 and 140**.
>
> ---
>
> - [a] Ozcelik, F. and VanRullen, R. (2023). Natural scene reconstruction from fMRI signals using generative latent diffusion. Nature Scientific Reports, 13.

---

> > ### Comment · Reviewer_eXwK · 2024-11-25
> >
> > Clarification regarding the possibility of an additional baseline: The table currently shows an MSE of 79.2 between the reference image and random images, versus 30 (approximately) for the targeted image variations. In your response, you seem to assume that SDEdit + ImageAdapter variations of the reference image (without brain-conditioned targeting) would give an MSE near zero, but my understanding is that diffusion-based image variations are not identical copies of the reference image, so I'm just asking what would be the correponding MSE. I think it makes a difference if it is, e.g., 20-30 (meaning that the targeted image variation is very close to an unconditional image variation) or if it is truly close to 0.

---

> > > ### Author Response · Authors · 2024-11-25
> > > **Computation of additional baselines**
> > >
> > > Thank you for this clarification. **We have added Appendix A.12 to tackle these points and illustrate the explanation below**:
> > >
> > > Since BrainACTIV injects brain-derived conditions (namely, the intermediate embeddings from equation 5) into the generation process using IP-Adapter, we assume that generation "without brain-conditioned targeting" means that we use IP-Adapter with $\alpha=0$ (equivalent to simply passing the reference image through IP-Adapter, as would be commonly done for generating image variations).
> > >
> > > Hence, for computing additional baselines, we take MSE (and LPIPS) between the reference image and the synthesized images using different values of SDEdit's $\gamma$ (i.e., $\alpha=0$ and $\gamma \in [0,0.1,...,0.9,1]$). Intuitively, the difference between these synthesized images is that $\gamma=1$ shows how CLIP "interprets" the semantic content in the reference, while $\gamma=0$ has no effect from CLIP and $\gamma$'s in-between are an interpolation between these two endpoints. Hence, all of these represent different interpretations of what would be considered "unconditional image variation" in the context of BrainACTIV. For example, Figure 32 values at $\gamma=1$ could be considered a lower baseline, while values at $\gamma=0.6$ could represent an upper baseline.
> > >
> > > For clarification: in our first response, we understood you referred to the specific case of $\alpha=0, \gamma=0$, which leads to metrics very close to zero. However, using the wider range of $\gamma$ indeed provides important context into the effect of brain conditioning on structural control w.r.t. unconditional image variation.

---

> > > > ### Author Response · Authors · 2024-11-25
> > > > **Clarification on a mistake for gamma=0**
> > > >
> > > > Our apologies. In the above reply, we made a mistake regarding $\gamma=0$: our code implementation simply passes the original image as a result (without using the diffusion model), hence the metrics (extremely) close to zero.
> > > > However, the rest of the arguments still hold if we instead look at $\gamma \approx 0$ (in this case, $\gamma=0.1$).

---

### Official Review · Reviewer_Dc87 · 2024-11-02

**Soundness:** 3
**Presentation:** 3
**Contribution:** 3
**Rating:** 8
**Confidence:** 4

**Summary:**

The paper introduces a new model BrainACTIV to generate synthetic images optimizing specific brain responses from specific areas. Their model incorporates recent works on diffusion models conditioning to generate images that vary structurally with brain signal for better interpretations of brain representations and downstream analysis.

**Strengths:**

The paper is novel in tackling a common interpretability problems faced among works of synthesizing image based on brain signal. These synthetic images usually requires a lot of guesswork to confirm they are representing information from the brain. The IP-Adaptor is a clever way to induce structural biases into diffusional models so that the generated image vary reliably on the axis that encode the most salient representation that a brain areas cares about. This model has great potential to be extended to different brain data modality and it seems like it only requires a CLIP based brain encoding model for adaptation.

The second part of the paper analyzing mid-level features provides good examples of potentially making inferences from these synthetically generated images, further making use of the interpretability of the method.

**Weaknesses:**

The paper should perhaps incorporate predictive performance the Dino-ViT and fwRF encoder of these brain areas in Figure 3b so reader could have a better sense of how big of a difference the manipulation of images could make.

My main critique of this paper is perhaps more high level. Now that we have a model that could generate very clean optimized images for specific brain areas, can one use it for making new discoveries/inferences about how the brain represent visual information? Most of the work that synthesize images with brain responses has been proving that model can generate what we already know about the brain (e.g. Face image with FFA data) but rarely do they inform what we don’t already know. I am aware this paper might want to establish itself as a proof that this method could work in generating testable hypothesis but I am not yet convinced that it is substantially different than all the other papers in this subfield.

**Questions:**

line 323: what's the reasoning behind choosing images with responses closest to baseline activation?
Table 1: it might help to show with image what you means by "preserve structural fidelity" since "preserved" can be a relative concept. It might even be helpful to compute the numbers for images synthesized from other papers for comparison.

---

> ### Author Response · Authors · 2024-11-24
> **Response to Reviewer Dc87**
>
> We thank **Reviewer Dc87** for the positive evaluation of BrainACTIV and the insightful questions regarding novel discoveries. We address these below, and incorporate the concrete suggestions into our paper.
>
> > ### **W1: Predictive performance**
>
> Thank you for this suggestion. We agree that the predictive performance of the encoders provides valuable context to interpret the predictions in Figure 3(B). Because of space constraints, we have included these performance metrics in Appendix subsection A.2. **Based on other reviewer comments, we have also updated this subsection to display the predictive performance of our CLIP encoder. Accordingly, we have updated Figure 3 with the corresponding CLIP encoder predictions**.
>
> > ### **W2: New discoveries**
>
> The discovery of novel or finer-grained stimulus representations and organizational principles in the visual cortex is a very important topic in our line of research, and one that must certainly be emphasized in follow-up studies through collection of brain responses to synthesized images in the MRI scanner. **We have included an example on the use of BrainACTIV to formulate hypotheses for less well-understood regions (particularly, anterior IT cortex) in Appendix A.7**.
>
> BrainACTIV stands out from current approaches to optimal visual stimulus generation in that our synthesized stimuli are directly comparable to a reference image, whose structural properties are preserved. The relevance of this contribution can be understood from different points of view:
>
> - The successful modulation of predicted brain responses through changes in semantic categories in the image (while controlling for low-level structure) enables a strong test (and evidence for) the (well-known) category selectivity theory (Figure 2).
> - BrainACTIV can be used by experimenters to produce controlled experimental stimuli, where the hypothesized tuning axis of a group of voxels is derived in a data-driven manner. Additionally we describe how researchers can select the degree of structural control and semantic variation (Section 4.5).
> - By manipulating a reference  image (instead of generating novel images from randomly sampled noise), the analyses required to formulate new hypotheses become significantly more efficient in terms of compute costs (less images are needed) and automatization (no need for human inspection of hundreds of images).
>
> Additionally, we present a novel way to generate fine-grained hypotheses about ROIs with similar category preferences in Section 4.4, by isolating what distinguishes one ROI from the other and directly accentuating it in an image. We believe this contribution can be used not only to explore selectivity in novel regions of interest, but also to expand our knowledge about well-known category-selective regions.
>
> > ### **Q1: Closeness to baseline activation**
>
> Thank you for the opportunity to clarify this choice. We select images closest to baseline activations primarily for efficiency purposes: because the ROI-specific modulation embeddings point in the same direction regardless of the image, the semantic categories that appear in the manipulations vary minimally across images. Hence, we avoid the manipulation of the entire test set (1,000 images) and instead focus on a smaller subset. Using images with activation close to baseline probabilistically represents an “average” visual stimulus (i.e. there are significantly more stimuli with baseline activation than with optimal activation) and intuitively provides more room for maximizing/minimizing activity. Because this choice could bias the selection (e.g., towards a single semantic category that always elicits baseline activation), we employed six mutually-exclusive image subsets to enforce the diversity of the 120-image set.
>
> > ### **Q2: Structural fidelity**
>
> Thank you for pointing out this ambiguity. **We have updated Table 1’s caption and line 361 to emphasize our goal**: reference and variations should be as similar as possible in spatial structure and color palette. Because relevant works (e.g., NeuroGen [a] and BrainDiVE [b]) do not use a reference image or control the low-level structure of their images, a direct comparison in this table is unfortunately not possible.
>
> ---
>
> - [a] Gu, Z. et al. (2022). NeuroGen: Activation optimized image synthesis for discovery neuroscience. NeuroImage, 247:118812.
> - [b] Luo, A. F., Henderson, M. M., Wehbe, L., and Tarr, M. J. (2023). Brain diffusion for visual exploration: Cortical discovery using large scale generative models. NeurIPS.

---

> > ### Comment · Reviewer_Dc87 · 2024-11-25
> > **Thank you for detailed response**
> >
> > Thank you for detailed responses.
> >
> > I went through Appendix A.7 and that somewhat validates what I was arguing. Obvious semantic categories can be easily pulled out with this method (component 1 for food for example). These areas very likely show up as well in published localizer experiments (e.g. food ROIs in Jain et al. 2023) and this method does well in validating them. For areas identified by component 2 and 4, it is still hard to interpret what they are coding for and whether there should be a semantic category label attached to it at all. However I do still think this is progress compared to what people are capable to do before with ROI localizer and encoding models.

---

### Official Review · Reviewer_82Gz · 2024-11-02

**Soundness:** 3
**Presentation:** 3
**Contribution:** 2
**Rating:** 5
**Confidence:** 4

**Summary:**

At its core, the methodology employs a CLIP-based brain encoder that transforms visual inputs into corresponding brain activation measured with fMRI, with the goal to generate interpretable images that excite or inhibit certain brain areas. The proposed method, BrainACTIVE, allows for incorporating semantic information from modulation embeddings as well as lower-level image information directly into the generation process by seeding a diffusion process specific target images. The authors validate their proposed model by training separate encoders to predict the brain's activity to the synthesized images.

**Strengths:**

The paper overall is very well written, clear, with a straightforward exposition and extensive experiements. Sufficient detail is provided to understand all key components of the data, model architecture, and training.

**Weaknesses:**

While the paper is very well presented, I'm taking issue with the related work section, the selection of experiments, and the overall novelty. Considered as a whole, the paper can be improved significantly by addressing these concerns.


**Major Concerns**
- *Related works*: there is a wealth of articles surrounding generative models for optimizing stimuli for neuronal data, especially using fMRI, see related literature (1, 2, 3, 4). ]The focus of the related work sections seems to lie on category selectivity in visual cortex and and diffusion models using CLIP. More emphasis on the wealth of related work could be placed to compare the proposed method to other approaches

- *Comparing predictive performance*: Having separate encoding models is a clear strength of this paper. However, the predictive performance of the CLIP encoder model was not convincingly shown. Showing that also the CLIP-based encoding model has a comparable performance on the held-out test set (as shown in Fig. 9) as the DINO and fwRF would provide valuable insight.  A similarly interesting analysis would be including the CLIP-encoding model into Fig. 3 to understand the change in neuronal activity from the generating model. Lastly, it is important to demonstrate how much of the variations in the images is due to the random seed in the generator models - by training different models and measuring the similarity of the optimal images, and by measuring the variability of activations in the evaluator models (DINO and fwRF)

- *Analyses of mid-level images features*: This analysis shows much promise but doesn't yield much insight. To establish the validity of these measures, it would have been important to show that the metrics are consistent for either handpicked or parametric stimuli. Similarly, these metrics could be very useful to study early- to mid-level ROIs.

- *Overall novelty*: Taken together, while this approach is promising, it doesn't deliver a new insight into either category selectivity of mid-level image feature processing of visual cortex. The analyses are mostly confirmatory with respect to the current state of understanding of visual processing. The authors could take one of the so called "no-mans-land" ROIs and discover novel categories that drive these regions, such as anterior IT.


Literature:
- (1) https://medarc-ai.github.io/mindeye

- (2) https://mind-vis.github.io

- (3)  https://arxiv.org/abs/2305.10135

- (4)  https://proceedings.neurips.cc/paper_files/paper/2023/hash/67226725b09ca9363637f63f85ed4bba-Abstract-Conference.html

**Questions:**

- The usefulness of the top-nouns analysis is unclear to me, and the authors don't discuss the outcome of these results in the discussion section. In general, applying BrainACTIV seems very promising when applying it to areas in the visual hierarchy that are less well understood, going beyond largely confirmatory analyses. It would be great if the authors could include further discussion points how the top-noun analysis could best be utilized in the discovery of new feature or category selectivities.

---

> ### Author Response · Authors · 2024-11-24
> **Response to Reviewer 82Gz (1/2)**
>
> We appreciate the concrete suggestions made by **Reviewer 82Gz**. We have carefully considered them and adapted our paper accordingly. We address each suggestion below and gladly welcome further questions about them.
>
> > ### **W1: Related works on stimulus reconstruction**
>
> Thank you for this important suggestion. Due to the wealth of work on both image reconstruction / brain decoding and data-driven exploration of category selectivity (particularly using the Natural Scenes Dataset), we initially decided to narrow the focus of our Related Work section to studies that specifically focus on optimal visual stimulus generation.
>
> For clarification, we would like to emphasize the two different directions regarding brain-conditioned image generation: the first one tackles (exact) reconstruction of observed stimuli based on the brain activity patterns they elicited ([a][b][c]) while the second one concerns the generation of novel stimuli that activates a specific brain region. These two directions share many components (particularly, those using diffusion models and fMRI data), and the methods can often be adapted to perform both of them (see [d][e]). We therefore acknowledge the vast amount of work performed on image reconstruction from brain activity, and its importance to our work. **Accordingly, we have adapted the Related Work section by adding the suggested citations**. Because our work pertains solely to the second mentioned direction, we preserve the focus of this section on works that deal with optimal visual stimulus generation aiming to provide better context on the evolution of this subfield of research.
>
> > ### **W2: Performance of CLIP encoder**
>
> We strongly agree that the performance of the CLIP encoder provides important context for our results. **Therefore, we have adapted Figure 9 to display the performance of the three brain encoders on the held-out test set for all subjects and ROIs. We have also added the corresponding CLIP encoder predictions to examples in Figure 3(A), as well as predicted modulation results from the CLIP encoder in Figure 3(B)**.
>
> Due to time constraints, it was not possible to provide a comprehensive analysis on the effect of random initialization of the encoders and generative model on our results. However, to provide further evidence that BrainACTIV minimizes the effect of random variability from the generator model, and better isolates the effect of brain optimality, **we have displayed example variations for FFA and PPA over multiple different seeds (for the diffusion model) in Appendix A.10**.
>
> ---
>
> - [a] Scotti, P. S. et al. (2023). Reconstructing the Mind’s Eye: fMRI-to-Image with Contrastive Learning
> and Diffusion Priors. NeurIPS.
> - [b] Chen, Z. et al. (2023). Seeing beyond the brain: Masked modeling conditioned diffusion model
> for human vision decoding. CVPR.
> - [c] Zeng, B. et al. (2023). Controllable mind visual diffusion model. arXiv preprint. arXiv:2305.10135.
> - [d] Ozcelik, F. and VanRullen, R. (2023). Natural scene reconstruction from fMRI signals using generative latent diffusion. Nature Scientific Reports, 13.
> - [e] Papale, P., De Luca, D., and Roelfsema, P. R. (2024). Deep generative networks reveal the tuning of
> neurons in IT and predict their influence on visual perception. bioRxiv, pages 2024–10.

---

> > ### Author Response · Authors · 2024-11-24
> > **Response to Reviewer 82Gz (2/2)**
> >
> > > ### **W3: Mid-level features and early ROIs**
> >
> > We agree that our analysis of mid-level features is limited in scope and requires further validation in future work. We would like to clarify that these four mid-level features are not meant to form a comprehensive analysis of the role of mid-level features representation in the human visual cortex, but rather to serve as an illustration of how BrainACTIV—because of its use of a reference image as baseline—can give precise quantitative metrics of the effects of brain optimization on images and on predicted brain activations. **We have updated Methods section 3.2 to better explain our intention here**, while we have also **edited Results section 4.3** to highlight how these findings connect with the existing literature, and that they should be taken as indicative of BrainACTIVs potential for new hypothesis generation, rather than definitive proof for mid-level feature representation in the brain. As additional context for these features, **we have included Appendix A.11 to illustrate how each of the features is represented at extreme values in NSD**.
> >
> > We also agree that these metrics (and the applicability of BrainACTIV in general) could be further illustrated by targeting early- to mid-level regions. **Accordingly, we have added Appendix A.6 to illustrate results on areas V1 through V4**. These results highlight BrainACTIV’s success in modulating predicted responses, as well as its limitations regarding the use of CLIP’s image space as a model of the target ROI. Namely, even though these regions are known for their selectivity to low- and mid-level properties, the use of a semantic image space causes the changes within the manipulations to be mainly high-level. Interestingly, however, these high-level changes can be interpreted as real-life occurrences of high- and low-level properties, for example, bright light -> light bulbs.  **We quantify this in Figure 19, where we observe stronger changes in features like color saturation and entropy for V1-V4 than for category-selective ROIs (Appendix A.8)**.
> >
> > > ### **W4 and Q2: Overall novelty and no-man’s-land**
> >
> > We appreciate the suggestion regarding the application of BrainACTIV to less-understood ROIs. We acknowledge that our method, together with many recent studies on data-driven investigation of neural representation, must emphasize the discovery of novel or finer-grained selectivity and organizational principles in the visual cortex. **Based on this suggestion, we have introduced Appendix A.7 with a brief example on how BrainACTIV can be adapted to investigate selectivity in anterior IT**.
> >
> > > ### **Q1: Top-nouns analysis**
> >
> > Thank you for raising this concern and for acknowledging BrainACTIV’s potential. The top-nouns analysis is meant to illustrate that BrainACTIV effectively discerns between regions that are selective to the same broad category (in this case, places → OPA/PPA/RSC). This distinction is visually apparent in the image manipulations, but not in our category analyses (Figure 4) since the 16 broad categories we choose in Section 3.2 are not sufficiently fine-grained. Hence, we use individual nouns (obtained from WordNet as outlined in Appendix subsection A.1), showing that the specific set of nouns highlighted for each region emphasizes differences in selectivity across them.  **We have changed the order of presentation of the results** so that the top-nouns analysis now accompanies the accentuation of differences between ROIs in Section 4.4 (hence in the new Figure 6).
> >
> > For additional clarification: we performed this analysis to show that BrainACTIV retains one of the main strengths of BrainDiVE [f]. Namely, characterizing differences between ROIs selective for the same high-level category. We later expand on this by contributing a way to isolate these differences and accentuating them through image manipulation. **We have updated the text (Section 4.4) to include the explanation above**.
> >
> > ---
> >
> > - [f] Luo, A. F., Henderson, M. M., Wehbe, L., and Tarr, M. J. (2023). Brain diffusion for visual exploration: Cortical discovery using large scale generative models. NeurIPS.

---

### Official Review · Reviewer_mur3 · 2024-11-06

**Soundness:** 3
**Presentation:** 4
**Contribution:** 3
**Rating:** 6
**Confidence:** 5

**Summary:**

In this paper, the authors present a method for generating maximizing or minimizing images for a brain ROI using Stable Diffusion. The method uses the NSD dataset, and builds on previous work using this dataset and diffusion models to generate activating images. Specifically, starting from a source image, the method changes this image to maximize or minimize the activation in an ROI (though the estimation of a CLIP vector characterizing the ROI from an encoding model). This makes it possible to detect the change in category representations as different regions are maximized. The method can also be used to contrast two ROIs, even ones that process the same category.

**Strengths:**

- The method nicely builds up on previous work (which it acknowledges well).
- It allows a more concrete evaluation of images across ROIs due to the presence of the common starting image.
- The formulation allows for both maximization and minimization.
- It is possible to contrast multiple ROIs along the same brain pathway.
- The results makes sense in terms of the discovered selectivities.

**Weaknesses:**

1- More quantification of the observed effects is needed, or at least more details about them.
2- More should be discussed in terms of the meaning of the maximizing/minimizing categories and the results. It is not clear what are novel  results about the brain and what is a replication.
3- The most novel results appear to be in figure 7. However these are not discussed in terms of significance or hypotheses.
4- Luo et al 2024 is used in the methods but doesn't figure in the related work section. How does the current work compare with the method used there?

**Questions:**

- Regarding Weakness 1: How many images are used to compute the selectivity results (e.g. figure 4), why are those chosen? What is the predicted activity associated with the maximizing and minimizing images? What are the statistics related to these activities? Could it be used to make predictions about how such images would activate or deactivate brain regions?
- Regarding Weakness 2: how do we understand the minimizing categories? Is it just a factor of having to step away from the preferred categories or is there truly a reduction in activity from baseline? Is this supported in the literature?
- Regarding Weakness 3: What hypotheses can be predicted from the results comparing two regions?

---

> ### Author Response · Authors · 2024-11-24
> **Response to Reviewer mur3 (1/2)**
>
> We thank **Reviewer mur3** for the positive comments and insightful questions. We have revised our paper based on these questions to provide more context into the relevance of our work.  We will address your questions and suggestions below.
>
> > ### **Quantification / details of the observed effects**
>
> ### *How many images are used to compute the selectivity results (e.g. figure 4), why are those chosen?*
>
> All analyses in Section 4.3 (specifically, Figure 4 and Figure 5) use the results from our main experiment (Section 4.2). Namely, 20 variations for each of 120 test images in each subject. To aggregate these results into a single figure, we average over all subjects and all images. We make this particular choice because the modulation embeddings z_max and z_min, used as endpoints to generate the subject- and ROI-specific image variations, are highly similar across subjects (please see the newly added Appendix subsection A.8). Hence, the semantic and mid-level features modified in the images vary minimally across subjects. **We have updated the first paragraph of Section 4.3 to clarify this information in the main text.**
>
> ### *What is the predicted activity associated with the maximizing and minimizing images?*
>
> Figure 3 (A) displays example results for our maximization and minimization experiment from Section 4.2. For each of them, we show the activity of the target region of interest as predicted by each of our three brain encoders (**based on the other reviews, we have updated Figure 3 to also display predictions from our CLIP encoder**). We further show the average of these values over all images and subjects (with a reasoning similar as mentioned above) in Figure 3 (B) to evidence BrainACTIV’s potential to modulate activity in these regions.
>
> ### *Could it be used to make predictions about how such images would activate or deactivate brain regions?*
>
> While the effect of our synthesized images must be verified in future work by presenting them to new subjects in an MRI scanner, the observation of a clear pattern across ROIs in Figure 3 (B), as well as the use of two different encoders (namely, DINO-ViT and fwRF) that are not based on CLIP,  provides robust support for BrainACTIV’s applicability in such online experimental settings. **We have also updated Figure 9 to display the predictive performance of these three encoders**, in order to provide further context to the predictions.
>
> > ### **Discussion about minimizing categories**
>
> ### *How do we understand the minimizing categories? Is it just a factor of having to step away from the preferred categories or is there truly a reduction in activity from baseline?*
>
> Here, minimization should be understood as the ROI’s level of activity relative to the reference image. Compared to the no-stimulation baseline, category-selective ROIs typically respond positively to all images (including non-preferred categories; see Fig 3 and 4 in [a] for examples), but much more strongly to the preferred category. In our analysis, we purposely chose reference images from NSD that yielded average activations for each ROI, so lying somewhere between the minimum and maximum for that ROI. BrainACTIV then exploits this range in activity to push the reference towards the tails of the distribution of (positive) activations for that ROI . Hence, our minimization should not be interpreted as ‘deactivating’ or suppressing fMRI activations, but rather as leveraging the full range of activation rather than only considering the top-activating images, as is typically done in MEI studies. **We have updated our phrasing in the Abstract, and main text to clarify this**.
>
> ### *Is this supported in the literature?*
>
> In terms of the observed results with minimization, these are broadly consistent with the known literature (e.g. face-selective regions respond the least to places and vice-versa), but some of the minimization effects we observe are, to our knowledge, not yet documented in the literature and could yield new hypotheses about these regions that can be tested in future studies. **We have updated the first paragraph of Section 4.3 to explicitly mention this in the main text**.
>
> ---
>
> - [a] Groen IIA, Silson EH, Pitcher D, Baker CI. (2021). Theta-burst TMS of lateral occipital cortex reduces BOLD responses across category-selective areas in ventral temporal cortex. NeuroImage (230) 117790 https://doi.org/10.1016/j.neuroimage.2021.117790

---

> > ### Author Response · Authors · 2024-11-24
> > **Response to Reviewer mur3 (2/2)**
> >
> > > ### **Hypotheses predicted from the comparison between regions**
> >
> > Figure 7 (now Figure 6) indeed demonstrates how BrainACTIV can be used to reveal novel insights about fine-grained differences between ROIs with the same category-selectivity. In section 4.4, we describe the apparent differences between images that are accentuated for each pair of ROIs with the same category selectivity. By having these accentuations embedded in the image generation process, BrainACTIV directly generates new hypotheses about the differences between the ROIs, which can then be tested in future experiments by using the generated images as stimuli. **We have changed the order of presentation of the results** so that Figure 7 now follows directly after Figure 5, highlighting the novelty of the hypotheses generated through the accentuation method, relative to the top-nouns analyses or ROI comparisons done in e.g. BrainDiVE [b]. **In addition, we have edited our phrasing in section 4.4. to explicitly note that we consider the observed differences as new hypotheses**.
> >
> > > ### **Comparison to BrainSCUBA (Luo et al., 2024)**
> >
> > Thank you for this important remark.
> >
> > Similar to our work, BrainSCUBA [c] attempts to solve a problem with current optimal (visual) stimulus generation methods (particularly,  BrainDiVE). Namely, the reliance on visual inspection of synthesized images to formulate new hypotheses about category selectivity, and their consequently low interpretability. Their method further employs a similar CLIP-based brain encoder as in our work. However, the main focus of their approach is on the generation of interpretable (natural language) captions that elucidate semantic selectivity properties at the voxel-level. While they additionally use these captions to prompt text-to-image diffusion models (yielding cleaner images than BrainDiVE), we consider that the methodological problems we point out in Section 2—independently generating one stimulus at a time—remain in BrainSCUBA. Moreover, we believe images still yield richer semantic and low-level information than natural language captions, and work on computer vision can be leveraged to quantify highly specific properties (as we attempt in Section 3.2). Finally, BrainACTIV additionally tackles the production of controllable stimuli for novel scientific experiments (**see revised Section 4.5**). Still, the work on BrainSCUBA has been really valuable for BrainACTIV’s implementation, and its alternative way to tackle current limitations in the field through analyses on natural language is worth investigating further.
> >
> > Because there is an increasingly large number of studies on data-driven investigation of cortical selectivity (particularly using the Natural Scenes Dataset), we decided to narrow the focus of our Related Work section to those that specifically focus on generation of novel visual stimuli that maximizes activity in target brain regions. **However, we have added a sentence to our Discussion highlighting how the adaptation of BrainACTIV’s manipulation approach to natural language could yield valuable insights on semantic selectivity**.
> >
> > ---
> >
> > - [b] Luo, A. F., Henderson, M. M., Wehbe, L., and Tarr, M. J. (2023). Brain diffusion for visual exploration: Cortical discovery using large scale generative models. NeurIPS.
> > - [c] Luo, A. F., Henderson, M. M., Tarr, M. J., and Wehbe, L. (2024). BrainSCUBA: Fine-grained natural language captions of visual cortex selectivity. ICLR.

---

### Author Response · Authors · 2024-11-24
**Summary of our response**

## We thank all reviewers for their helpful and constructive comments.

We were pleased to see that all reviewers considered the soundness and presentation of our paper to be good to excellent, highlighting its clarity, innovativeness and potential for new neuroscientific discovery. In our revision, we implement substantial changes that we hope will better convey the novelty of BrainACTIV and its value as a scientific contribution. Since some comments were shared across multiple reviewers, we highlight the most significant revisions here:

- To stress the value of our novel method of image variation for neuroscientific hypothesis testing, we have reordered and expanded previous Results sections 4.3 to 4.5, resulting in a more logical flow from reproducing established findings to novel insights gained by BrainACTIV.
- We have expanded prior literature to better acknowledge decoding/reconstruction literature that BrainACTIV builds upon.
- We now report CLIP encoder performance in the main text Figure 3 and Appendix A.2.
- To showcase how BrainACTIV can generate novel insights beyond already-known category-selective regions, we have added additional analyses of early visual and anterior IT ROIs in Appendix A.6 and Appendix A.7, respectively.
- Also, while not explicitly requested by any reviewer, we replaced the original projection set for a much larger one (line 302) to reduce bias in the results, so we updated all figures accordingly.

We provide more specific responses to all reviewer comments below. We are hopeful that with these revisions, all reviewers will achieve consensus on the valuable contribution of BrainACTIV.

---

### Meta-Review · Area_Chair_4Wp6 · 2024-12-17

**Metareview:**

This paper introduces BrainACTIV, a method for generating images that optimize brain activation in specific regions of interest (ROIs) by manipulating a reference image using diffusion models. The approach allows for both maximizing and minimizing brain activity, offering a new way to contrast and analyze brain regions' selectivity for different categories.

Weaknesses identified in the review, such as the need for more quantification of the effects, clearer discussions on the significance of results, and further exploration of comparisons to related methods, were mostly addressed during the authors' rebuttal.

The paper's strength lies in its novel approach to brain region-specific image manipulation and its potential to refine how we understand brain activation. Therefore, I recommend accepting the paper for its innovative methodology and valuable contributions to neuroscience.

**Additional Comments On Reviewer Discussion:**

The authors addressed multiple reviewer concerns regarding the BrainACTIV methodology, providing clarifications and additional details. They explained that the minimization process explores a wider range of activations relative to a reference image, rather than deactivating brain regions, and suggested that it could generate new hypotheses for future experiments. They also emphasized how BrainACTIV accentuates differences between regions with similar category selectivity, facilitating hypothesis generation. The authors clarified the performance of the CLIP encoder, incorporating comparisons in updated figures, and addressed the analysis of mid-level features and early visual cortex regions by explaining their role in illustrating the tool's potential. They provided more context for the top-nouns analysis, which helps highlight differences between regions selective for the same category. Further, the authors justified using images closest to baseline activations for efficiency and explained the rationale behind baseline comparisons, including an appendix with additional baseline computations. Regarding the novelty of BrainACTIV, they clarified that the integration of existing methods with novel applications in neuroscience, such as IP-Adapter and SDEdit, represents a significant computational advancement. I consider these comments to be the most important and they were all addressed by the authors.

---

### Decision · Program_Chairs · 2025-01-22

Accept (Poster)